# Advances in the Synthesis of Biologically Active Quaternary Ammonium Compounds

**DOI:** 10.3390/ijms25094649

**Published:** 2024-04-24

**Authors:** Joanna Fedorowicz, Jarosław Sączewski

**Affiliations:** 1Department of Chemical Technology of Drugs, Faculty of Pharmacy, Medical University of Gdańsk, Al. Gen. J. Hallera 107, 80-416 Gdańsk, Poland; 2Department of Organic Chemistry, Faculty of Pharmacy, Medical University of Gdańsk, Al. Gen. J. Hallera 107, 80-416 Gdańsk, Poland; js@gumed.edu.pl

**Keywords:** analgesics, anticancer drugs, antibacterial agents, anticholinoesterases, antifungal compounds, antiprotozoals, antivirals, disinfectants, herbicidals, quaternary ammonium salt

## Abstract

This review provides a comprehensive overview of recent advancements in the design and synthesis of biologically active quaternary ammonium compounds (QACs). The covered scope extends beyond commonly reviewed antimicrobial derivatives to include synthetic agents with antifungal, anticancer, and antiviral properties. Additionally, this review highlights examples of quaternary ammonium compounds exhibiting activity against protozoa and herbicidal effects, as well as analgesic and anesthetic derivatives. The article also embraces the quaternary-ammonium-containing cholinesterase inhibitors and muscle relaxants. QACs, marked by their inherent permanent charge, also find widespread usage across diverse domains such as fabric softeners, hair conditioners, detergents, and disinfectants. The effectiveness of QACs hinges greatly on finding the right equilibrium between hydrophilicity and lipophilicity. The ideal length of the alkyl chain varies according to the unique structure of each QAC and its biological settings. It is expected that this review will provide comprehensive data for medicinal and industrial chemists to design and develop novel QAC-based products.

## 1. Introduction

Quaternary ammonium compounds (QACs) feature a distinct molecular structure characterized by a central nitrogen atom bonded to four organic groups and a negatively charged ion, typically a halide or other organic anion. This arrangement imparts a permanent positive charge to the nitrogen atom, making QACs cationic in nature (NR_4_^+^). The four organic groups attached to the central nitrogen can vary widely, ranging from alkyl or aryl chains to more complex structures. Their amphiphilic nature allows them to exhibit both hydrophilic and hydrophobic properties, enabling effective interactions with polar and nonpolar molecules. The structure of QACs grants them significant versatility and utility across various applications. Among others, they can be used as anti-infectives such as antiseptics for wounds and preoperative unbroken skin preparation, disinfectants of medical instruments and apparatus or surfaces, preservatives and additives in drugs, corrosion inhibitors, as well as surfactants and detergents [1]. They are utilized in a variety of industrial and consumer products in chemistry (phase-transfer catalysts, ionic liquids, and low vapor pressure solvents), agriculture (herbicides and pesticides) [2,3], and in households (personal care products, sanitizers and cleansers, fabric softeners, and antistatic shampoos) [4,5]. Prolonged exposure to QACs has been linked to various health and environmental risks. Research indicates that QACs can accumulate in the environment, posing threats to aquatic organisms and ecosystems [6,7]. Moreover, the presence of QACs in municipal sewage sludge highlights their extensive commercial use and underscores the need for further research into their fate and toxicity [8,9]. Health risks associated with QACs include respiratory issues, skin irritation and sensitization, eye irritation, and allergic reactions [10,11,12].

In this review, which encompasses the original literature published from 2017 to 2023, the latest reports on the development of synthetic procedures for small-molecule QACs, along with their bioactivity, are presented. The modification of an active agent by quaternary ammonium (QA) group addition has been shown to enhance water solubility and biological activity [13,14,15]. Present and prospective trends in the application of QACs, including their utilization in various fields such as healthcare, agriculture, and industry, are also discussed, highlighting their potential impact on future research and development efforts.

## 2. Antibacterial Activity

QACs are important anti-infective agents. Their spectrum of efficacy covers microorganisms (bacteria, fungi, and algae) as well as viruses bearing lipid envelopes (such as SARS-CoV-2). QACs are able to penetrate rigid polymicrobial biofilm structures formed by many pathogenic bacteria and microscopic fungi that are resistant to conventional treatment. They are recommended by disinfection programs and good practice guidelines as the most appropriate first-line defense agents against nosocomial infections within hospitals to minimize the prescription of antibiotics [16,17]. Due to their low price, biocidal properties, and versatile uses, the most commonly applied QACs are well-established benzalkonium chlorides (BAC), cetrimonium bromide and chloride, as well as didecyldimethylammonium and dimethyldioctadecylammonium chlorides [18].

The accepted mode of antimicrobial action assumes non-receptor interactions and is related to opposite charges electrostatic attraction and hydrophobic contacts between negatively charged lipid cell membranes and positively charged nitrogen atoms [19,20] as well as lipophilic substituents of amphiphilic surfactants [19]. In the initial phase, cationic QA molecules are hypothetically attracted to phospholipid components of the cytoplasmic membrane; thereby, a net positive charge is introduced and membrane distortions are produced on depolarized microbial cells [21,22]. Divalent magnesium and calcium cations stabilizing the membrane surface are replaced by cationic QA molecules, decreasing membrane fluidity. Next, hydrocarbon tail domains present in the molecule penetrate the inner part of the bilayer membrane, then insert and integrate into the membrane structure due to their favorable partition. This process leads to membrane segmentation by the formation of transient pores or channels that increase permeability and result in the alteration of the physical properties of the membrane. The perforation progressively causes leakage of low molecular content from intracellular space, an outflow of the potassium ions along with other cytoplasmic constituents (i.e., proteins and nucleic acids), and disturbance of the biochemical processes inside the cell [22,23,24,25,26,27]. As a result, the disintegrated and solubilized membrane completely loses the physiological barrier and osmoregulatory functions. Consequently, proteins and nucleic acids are degraded, while the autolysis pathway is activated, which ultimately leads to the death of the cells with the destroyed structural organization [22,26,27].

QACs are usually more active against Gram-positive (G+) bacteria since this kind of microorganism possesses a single phospholipid cellular membrane and a thick multi-layer peptidoglycan cell wall, while Gram-negative (G−) strains are enveloped by a complex structure of a thinner peptidoglycan layer and two membranes, inner and outer. The external membrane is composed of proteins and lipopolysaccharides (LPS), which render the entrance of antibacterials problematic [26,28]. Mammalian cells are less susceptible to the toxic effects of QAC than bacteria because of differences in lipid composition. In detail, eukaryotic membranes are rich in zwitterionic lipids, such as phosphatidylcholines, while bacterial cells are enveloped in lipid layers containing a greater amount of negatively charged anionic lipids [29,30]. As a result, the membrane potential in eukaryotes is more positive compared to bacteria and the adsorption of QACs predominantly occurs on prokaryotic membranes. Lipophilicity is the essential parameter that determines the antimicrobial activity of the QAC, although it seems to be not an exclusive factor. Usually, the antibacterial properties of a molecule increase with the growth of the logarithm of the partition coefficient (logP) up to the cut-off value, then drastically decrease. This effect is associated with a disturbance of the hydrophilic–hydrophobic balance of a compound. Infiltration and transport of highly hydrophobic substances that are poorly water-soluble through the membrane can be hampered [31].

The synthesis of novel QACs with antibacterial properties is a popular topic recently undertaken by many research groups [32,33,34]. Table 1 and Table 2 summarize the values of minimum inhibitory concentration (MIC) of the synthesized QACs found in the literature.

**Table 1 ijms-25-04649-t001:** MIC values [µM] of the selected newly synthesized QACs against most common bacterial strains.

Compound	*S. aureus*	MRSA	VRE	*E. coli*	*K. pneumoniae*	*P. aeruginosa*	*A. baumannii*	*S. epidermidis*	*M. luteus*	*Pseudomonas* sp.	*B. subtilis*	*B. cereus*	*E. faecalis*	Ref.
**1b**	1.47	1.95	1.95	5.86	7.81	>500	3.91	-	-	-	-	-	-	[35]
**2c**	0.5	4	8	32	32	>1024	-	2	-	-	-	-	-	[36]
**4d**	50	-	-	230	-	-	-	-	70	1010	-	-	-	[37]
**5c**	80	-	-	230	-	-	-	-	80	2380	-	-	-	[38]
**6c**	-	-	-	-	-	12.7	-	-	-	-	12.7	-	-	[31]
**10**	8	-	-	16	-	63	-	-	-	-	-	-	-	[39]
**11c**	15.6	-	-	>125	-	-	-	-	-	-	-	-	-	[40]
**14c**	2.6	-	-	27.9	-	231.1	-	-	-	-	-	-	-	[41]
**15**	10	-	-	40	4	30	-	80	-	-	-	2	-	[42]
**16a**	2.44	-	-	19.5	19.5	78	-	1.83	-	-	-	1.22	-	[43]
**19**	39.1	-	-	-	-	625	-	-	-	-	-	-	-	[44]
**20**	3.55	7.09	-	454	227	-	-	-	-	-	-	-	-	[45]
**22**	1.71	1.71	-	874	874	>874	-	-	-	-	-	-	-	[46]
**23**	8.76	8.76	-	70.1	35.1	280	-	-	-	-	-	-	-	[46]
**34c**	0.6	-	-	5.5	-	-	-	-	-	-	-	-	-	[47]
**38**	0.25	0.5	-	2	-	8	-	-	-	-	-	-	4	[48]
**48**	17	-	-	74.7	-	-	-	-	-	-	-	-	-	[49]
**50d**	2	1	-	1	-	4	-	-	-	-	-	-	4	[50]
**51c**	1	1	-	2	-	8	-	-	-	-	-	-	8	[51]
**52a–c**	>250	>250	-	>250	-	>250	-	-	-	-	-	-	>250	[52]
**60d**	1	-	-	1	-	-	-	-	-	-	-	-	-	[53]
**62d**	6.1	-	-	24.4	-	97.6	-	-	-	-	24.4	-	-	[54]
**63e**	3.7	-	-	7.3	-	-	-	-	-	-	-	-	-	[54]
**64l**	0.6	-	-	5	-	5	-	-	-	-	0.6	-	-	[55]
**65d**	5–10	-	-	40	-	-	-	-	-	-	-	-	-	[56]
**66d**	0.8	-	-	1.6	-	-	-	-	-	-	-	-	-	[57]
**76e,g**	6.25	-	-	-	-	6.25	-	-	-	-	-	-	-	[58]
**79**	30	-	-	16	-	250	-	-	-	-	-	-	-	[59]
**80a**	160	-	-	80	-	50	-	-	-	-	120	-	-	[60]
**83c**	0.98	0.98	1.95	3.91	7.81	-	1.95	-	-	-	-	-	-	[61]
**103a–c**	-	-	-	>200	-	-	-	-	-	-	>200	-	-	[62]

- not tested.

**Table 2 ijms-25-04649-t002:** MIC values [µg/mL] of the selected newly synthesized QACs against most common bacterial strains.

Compound	*S. aureus*	MRSA	VRE	*E. coli*	*K. pneumoniae*	*P. aeruginosa*	*A. baumannii*	*S. epidermidis*	*M. luteus*	*Pseudomonas* sp.	*B. subtilis*	*B. cereus*	*S. typhimurum*	*E. faecium*	*E. faecalis*	*S. pneumoniae*	*S. pyogenes*	*S. mutans*	*P. vulgaris*	*α-H-tococcus*	Ref.
**3c**	250	500	-	-	-	-	-	-	-	-	-	-	-	-	-	-	-	-	-	-	[63]
**4d**	20	-	-	100	-	-	-	-	30	450	-	-	-	-	-	-	-	-	-	-	[37]
**5c**	30	-	-	90	-	-	-	-	30	950	-	-	-	-	-	-	-	-	-	-	[38]
**9**	2.5	-	-	12.5	-	-	-	-	-	-	-	-	-	-	-	-	-	-	-	-	[64]
**13c**	1	-	-	8	-	64	-	-	-	-	-	-	-	-	-	-	-	-	-	-	[65]
**14c**	2	-	-	16	-	128	-	-	-	-	-	-	-	-	-	-	-	-	-	-	[41]
**17**	-	>63	-	2	-	1	8	-	-	-	-	-	-	-	-	-	-	-	-	-	[66]
**18a**	0.5	1	-	-	-	-	-	-	-	-	-	-	-	63	4	-	-	-	-	-	[67]
**18b**	0.15	0.3	-	-	-	-	-	-	-	-	-	-	-	1.2	0.15	-	-	-	-	-	[67]
**24e**	-	-	-	-	-	-	-	-	-	-	-	-	-	-	>64	0.5	1	-	-	-	[68]
**25g**	25	-	-	>200	-	-	-	-	-	-	50	-	-	-	100	-	100	25	-	-	[69]
**26**	-	4	-	-	-	-	-	-	-	-	-	-	-	-	-	-	-	-	-	-	[70]
**27**	1	-	-	4	-	32	-	-	2	-	2	-	-	-	-	-	-	-	-	-	[71]
**28**	1	-	-	8	-	>64	-	-	2	-	1	-	-	-	-	-	-	-	-	-	[71]
**29b**	0.5	2–16	-	2	-	>64	-	0.5	0.5	-	-	-	0.5	-	-	-	-	-	-	-	[72]
**30**	1	-	-	16	4	32	-	2	-	-	4	-	-	-	-	-	-	-	-	-	[73]
**33b**	4	-	-	16	>64	64	-	4	-	-	4	-	-	-	-	-	-	-	-	-	[74]
**35g**	0.97	-	-	3.9	15.6	31.25	7.8	-	-	-	-	-	31.25	-	0.97	-	0.06	-	-	-	[75]
**36b**	0.4	-	-	15.6	-	500	-	-	-	-	-	0.9	-	-	-	-	-	-	-	-	[76]
**37a**	15.6	-	-	-	-	-	-	-	-	-	-	-	-	-	-	-	-	-	-	-	[77]
**43**	32	-	-	16	-	-	-	-	-	-	8	-	-	-	-	-	-	-	-	-	[78]
**44**	0.48	-	-	15.6	-	-	-	-	-	-	-	0.98	-	-	-	-	-	-	-	-	[79]
**45a–c**	32	-	-	64	-	-	-	-	-	-	-	-	-	-	-	-	-	-	-	-	[80]
**45d,e**	<2	-	-	256	-	-	-	-	-	-	-	-	-	-	-	-	-	-	-	-	[80]
**53b**	-	0.98	-	50	31	10.7	-	-	-	-	-	-	-	-	-	-	-	-	-	-	[81]
**54**	-	0.25	-	0.5	2	4	2	-	-	-	-	-	-	-	-	-	-	-	-	-	[82]
**55d**	-	-	-	16	-	-	-	-	-	-	-	-	-	-	-	-	-	-	-	-	[83]
**58**	-	-	-	4	-	-	-	-	-	-	-	-	-	-	-	-	-	-	-	-	[84]
**59b**	-	-	-	2	-	-	-	-	-	-	-	-	-	-	-	-	-	-	-	-	[85]
**65d**	4–8	-	-	32	-	-	-	-	-	-	-	-	-	-	-	-	-	-	-	-	[56]
**69b**	-	-	-	-	-	-	-	-	-	-	-	-	-	-	-	-	-	3.12	-	-	[86]
**70g**	2.5	-	-	2.5	-	-	-	-	-	-	-	-	-	-	-	-	-	2.5	-	-	[87]
**71d**	61.3	-	-	-	-	-	-	-	-	-	-	15.6	-	-	-	-	-	-	-	-	[88]
**73**	1	-	-	8	32	16	-	1	-	-	2	-	-	-	-	-	-	-	-	-	[89]
**75d**	2	-	-	0.25	-	1	-	1–4	-	-	0.5	-	-	-	-	512	-	-	0.125	-	[90]
**77a**	6.25	-	-	25	-	12.5	-	-	-	-	-	-	-	-	-	-	-	-	25	6.25	[91]
**77b**	6.25	-	-	12.5	-	25	-	-	-	-	-	-	-	-	-	-	-	-	12.5	6.25	[92]
**81**	<90	-	-	<90	-	-	-	-	-	-	-	-	-	-	-	-	-	-	-	-	[93]
**86d**	64	-	-	512	512	1024	-	-	-	-	-	-	-	-	-	-	64	-	-	-	[94]
**93a**	1.25	-	-	2.5	-	0.625	-	-	-	-	1.25	-	-	-	-	-	-	-	-	-	[95]
**94**	-	-	-	64	-	-	-	>16	-	-	-	-	-	-	-	-	-	-	-	-	[96]

- not tested.

### 2.1. Benzalkonium Analogues

BAC is probably the most commonly used QA disinfectant worldwide, for example, as a key component of Lysol. Its chemical structure comprises two methyl, benzyl, and variable long-chain alkyl (generally 8–18 carbons long) substituents of ammonium head. It is formed by the alkylation of dimethylbenzylamine with a variety of chloroalkanes (in fact, typically alkyl chlorides mixture is used) [26]. Compounds of this type were an inspiration for many scientific groups to develop novel cationic surfactants. For example, Marek and coworkers obtained BAC analogues by a nucleophilic substitution reaction of benzyl bromide and double long-chained tertiary amines (Figure 1). The compounds showed a high efficacy against the tested G+ and G− bacteria, similar to that of the standard BAC (Table 1). Lipophilicity of the compounds seemed to be a key factor for the determination of their biological activity, and derivative **1b** exhibited the highest antimicrobial potency. Unfortunately, a correlation trend of a cytotoxicity level with the increase in the carbon chain length was confirmed as expected. The compounds did not surpass their BAC analogues concerning their safety potential and selectivity for microbes over Chinese hamster ovary CHO-K1 mammalian cells. Therefore, they can be applied as disinfectants rather than antiseptics [35].

In the further study, benzoxonium-like salts differing in the length of alkyl substituent (C10–C18) were synthesized in a two-step reaction process (Figure 2). The compounds were tested against eight planktonic (4 G+ and 4 G−) and one biofilm-forming bacteria. The more lipophilic **2c**,**d** (C14,C16) derivatives were the most active against G+ strains, whereas G− organisms were more susceptible to slightly less hydrophobic **2b**,**c** (C12,C14) QACs (Table 1). Cytotoxicity determined in the CHO-K1 cell line indicates that the obtained compounds are generally safe and selective; therefore, they can be potentially used in practice [36].

Jadhav et al. designed and prepared a new class of heterocyclic QACs by condensation of 2-morpholinoethan-1-amine with a set of fatty acids, followed by quaternization with benzyl bromide (Figure 3). The formed morpholinium compounds incorporating a cyclized ‘ether bridge’ were found to be non-toxic and biologically safe for adenocarcinomic human alveolar basal epithelial cells A549, human embryonic kidney HEK 293, and human liver hepatocellular carcinoma HepG2 cell lines up to the highest concentration tested (100 µg/mL). Derivatives **3b** and **3c** bearing unsaturated hydrocarbon substituents exhibited antibacterial properties, with the best effective compound **3b** displaying a broad spectrum of activity against all G+ and G− bacterial strains used in the study. Nevertheless, the determined MICs were relatively high, in the range of 250–500 µg/mL (Table 2) [63].

Guastavino and coworkers designed and prepared biodegradable surfactants bearing QA moiety by benzylation or methylation of tertiary amino alkylo carbonate derivatives (Figure 4). The compounds were designed as environmentally friendly low-toxic “green surfactants” that can be made from renewable materials according to green chemistry principles. The novel cleavable carbonate compounds were screened against two G+ and two G− strains as well as two yeasts and two molds. **4c**,**d** and **5b**–**d** analogues exhibited the strongest activity against the tested microorganisms, with the best active compounds, **4d** and **5c**, bearing 12 and 10 carbon atom chains, respectively. The **4d** derivative showed MICs comparable to those of BAC (Table 1 and Table 2) [37,38].

Three series of 1,2,3-triazole-based QACs substituted with hydrophobic chains of various lengths were synthesized by Mechken et al. Figure 5 outlines the synthetic strategy based on amine alkylation with propargyl bromide followed by click 1,3-dipolar cycloaddition of azides and alkynes in the presence of copper ions. The obtained QACs were tested against G− and G+ bacterial and fungal strains. The results indicated that biological activity strongly correlates with the length of the hydrocarbon chain and the polar head substituent group. Firstly, the MIC values reach a minimum in the range of 12–14 carbon atoms in chain, then increase gradually. Secondly, the benzyl substituent enables enhancement of antimicrobial activity similarly to benzalkonium compounds. Benzyl derivative **6c** comprising a 12-carbon hydrophobic chain exhibited broad-spectrum activity against all tested organisms with low MIC values of 6.3, 12.7, and 12.7 µM for *Aspergillus niger*, *Pseudomonas aeruginosa*, and *Bacillus subtilis*, respectively. It is noteworthy that the fungal strain *A. niger* is usually very difficult to eradicate due to its resistance to classical chemical disinfectants. Overall, compounds of set **6** showed the best antimicrobial results. The authors suggest that it can be caused by a lack of steric hindrance, which is the reason why series **8** compounds are not active—the long alkyl chain prevents surfactant adsorption on the microorganism membranes [31].

Wang and collaborators obtained a ferrocene derivative of BAC, compound **9**, as an electrically switchable bactericide for spatial and temporal control of microorganisms (Figure 6). They hypothesized that the antibacterial activity of the designed QAC can be modulated by a reversible redox-induced transition between two states: the oxidized hydrophilic Fe^+^ form is formed as a free molecule in solution, while the reduced hydrophobic Fe species likely assemble into micelles. The growth of G− *Escherichia coli* and G+ *Staphylococcus aureus* was successfully inhibited at MICs of 12.5 and 2.5 μg/mL by QAC in oxidized and redox forms, respectively. The reported study proves that the proposed ferrocene QAC may establish a smart antibiotic for biomedical applications [64].

### 2.2. QA Peptidomimetics: Amino Acids, and Peptides

Antimicrobial peptides (AMPs) have been proven to be effective agents targeting bacterial membranes. Usually, they are lipophilic cationic oligomers rich in arginine and lysine (sometimes histidine) as well as hydrophobic amino acids. The structure of this important species was a starting point for the design of many novel QACs. Kumar and coworkers synthesized small molecular peptidomimetics via the ring-opening reaction of *N*-sulfonylisatins and primary amines, followed by conversion into QA iodides **10** (Figure 7). The compounds were obtained in good yields and without the requirement of any chromatographic purification. The antibacterial activity of the *N*-sulfonylphenyl glyoxamide-based AMP mimics was evaluated by in vitro assays against *S. aureus* and *P. aeruginosa*. None of the compounds were active toward G− strains and MIC values against G+ bacterium were relatively high, ranging from 63 to above 250 µM. Despite relatively weak antibacterial potency, the new antibacterial agents presented an acceptable therapeutic window. Cytotoxicity evaluated against normal human lung fibroblasts MRC-5 indicates that the compounds are safe since the half-maximal inhibitory concentration (IC_50_) determined for compound **10** (R^1^ = Br, R^2^ = octyl**)** was 328 µM, which gives a selectivity index (SI) of 5.21 in relation to MIC of 63 µM [97]. In the second step of the study, structure modification was introduced in order to widen the spectrum of activity, and various substituted isatin derivatives were used as substrates. The biphenyl backbone in the R^1^ position was found to be important for effectiveness against G− strains. Chloro-substituted QA iodide salt **10** (R^1^ = *p*-Cl-Ph, R^2^ = octyl**)** was identified as the most potent analogue, which shows excellent antibacterial activity against both G+ and G− bacteria (Table 2, MICs of 8, 16, and 63 μM against *S. aureus*, *E. coli*, and *P. aeruginosa*). This biphenylglyoxamide-based derivative disrupted 35% of preformed *S. aureus* biofilm at 32 μM [39], which was comparable to LL-37, a natural peptide in the II phase of clinical trials [98]. Tethered bilayer lipid membrane, as well as cytoplasmic membrane, permeability studies suggested that this compound exerts antibacterial action by permeation, depolarization, and disruption of bacterial membranes in a time- and concentration-dependent manner. In addition, the potent compounds were found to be non-toxic against mammalian cells at therapeutic dosages as evidenced by in vitro toxicity studies [39]. Subsequently, a library of amphiphilic anthranilamides **11a**–**e** as AMP mimics was synthesized by Kumar’s lab (Figure 7). The obtained AMP mimics were not active against *E. coli*, although their activity against *S. aureus* was fair (MIC 15.6–125 µM) (Table 1). Compound **11c** showed no cytotoxicity against human cells up to 50 µM [40].

Perinelli et al. proposed biocompatible surfactants **12a**–**c** and **13a**–**c** derived from the amino acids leucine (Leu), and methionine (Met), respectively, esterified with alcohols of different lengths (**a**—C10, **b**—C12, and **c**—C14) as presented in Figure 8. Their antimicrobial activity was tested on G+ (*S. aureus* and *Enterococcus faecalis*) and G− (*E. coli* and *P. aeruginosa*) bacterial strains as well as yeast (*Candida albicans*). In turn, the cytotoxicity effect was evaluated on the human colorectal cancer Caco-2 and human lung adenocarcinoma Calu-3 cell lines. Overall, no remarkable changes were detected in terms of MIC values between compounds with the same polar head (Leu or Met), which indicates that the kind of amino acid does not impact the activity of from the microorganism. Instead, biological activity was strongly dependent on the chain length, since MIC and IC_50_ values decrease with the increase in hydrophobicity. The surface-active antimicrobials **12b**,**c** and **13b**,**c** bearing 12- or 14-carbon chains, displayed MIC and IC_50_ values similar to those of BAC used as a reference compound (Table 2) [65]. In the next step, the authors synthesized BAC analogues **14a**–**c** based on leucine C10, C12, and C14 esters by the introduction of benzyl moiety (Figure 8). The toxicological profile of the obtained molecules was comparable to BAC in terms of cytotoxicity and hemolytic activity; however, no increase in antimicrobial activity was recorded (Table 1, Table 2, Table 3 and Table 4) [41]. The reported studies highlights the crucial role of the lipophilic tail and its influence on physicochemical and biological properties.

**Table 3 ijms-25-04649-t003:** MIC values [µM] for the selected newly synthesized QACs against most common fungi.

Compound	*C. albicans*	*C. tropicalis*	*A. niger*	*P. chrysogenum*	Ref.
**4d**	-	90	1240	-	[37]
**5c**	-	50	880	-	[38]
**6c**	-	-	12.7	-	[31]
**14c**	13.9	-	-	-	[41]
**34c**	0.6	-	-	-	[47]
**60e**	0.2	-	-	-	[53]
**62e**	12.2	-	97.6	48.8	[54]
**63e**	14.6	-	117	58	[54]
**64t**	8.2	-	33.2	16.5	[55]
**65d**	40	-	-	-	[56]
**66d**	1.6	-	-	-	[57]
**83c**	-	-	-	100	[61]

- not tested.

**Table 4 ijms-25-04649-t004:** MIC values [µg/mL] for the selected newly synthesized QACs against most common fungi.

Compound	*C. albicans*	*C. glabrata*	*C. tropicalis*	*A. niger*	Ref.
**4d**	-	-	40	550	[37]
**5c**	-	-	20	350	[38]
**13c**	2	-	-	-	[65]
**14c**	8	-	-	-	[41]
**33b**	0.38	-	-	-	[74]
**35g**	0.06	-	-	0.48	[75]
**36c**	2	-	-	500	[76]
**55d**	4	-	-	-	[83]
**65d**	32	16	-	-	[56]
**66d**	1.6	-	-	-	[57,99]
**77a**	6.25	-	-	-	[91,92]
**77b**	6.25	-	-	6.25	[91,92]
**86d**	64	-	-	-	[94]
**87**	625	>5000	2500	-	[100]
**88c**	1.6	-	-	-	[101]
**90a**	1	1	2	-	[102]

- not tested.

Met QA salt **15** was synthesized by Laulloo and coworkers via esterification of methionine with dodecyl alcohol followed by exhaustive methylation (Figure 9). The sulfur amino acid-based QAC displayed moderate to high antibacterial activity on G+ and G− bacteria with MIC values ranging from 2 to 80 µM (Table 1), similar to that of tetracycline used as a reference compound. Moreover, this compound was found to prevent G− *Klebsiella pneumonie* biofilm formation at 20 µg/mL. Detailed phospholipid binding studies revealed that the antibacterial action of the compounds could be attributed to a combination of hydrophobic and electrostatic interactions. In addition, the compound strongly binds to bovine serum albumin via hydrogen bonding, van der Waals’ forces, and hydrophobic interactions, which might be helpful in the distribution of the active substance within the body while used in treatment [42]. Laulloo’s research group also quaternized phenylalaninyl-proline dipeptide dodecyl and tetradecyl esters in order to prepare candidates for active ingredients in body wash formulations (Figure 9). The obtained QACs **16a**,**b** were evaluated against three G+ and four G− bacterial strains. As expected, the activity was higher towards G+ organisms. Due to its better solubility, the dodecyl derivative **16a** was more effective, with MIC values in the ranges of 1.22–2.44 and 19.5–78 µM towards G+ and G− bacteria, respectively (Table 1). Cytotoxicity determined in cervical adenocarcinoma HeLa and human foreskin fibroblasts Hs68 revealed that this compound is nontoxic and selective towards bacterial rather than mammalian cells only when applied at concentrations of MIC values for G+ strains since its IC_50_ values for corresponding cell lines were 17.09 and 42.19 µM, respectively [43].

Ongwae et al. synthesized 14 new polymyxin-based QA agents **17** type by solid-phase peptide synthesis and quaternization with lipophilic alkyl substituents on Wang resin (Figure 10). A basic scaffold of polymyxins contains two primary parts: a cationic lactam ring composed of seven amino acid residues and an exocyclic tripeptide with an acyl tail attached to the N-terminus. These molecules target LPS present in particularly difficult-to-eliminate G− bacteria. Novel structures retained the capability to target G− bacteria (Table 2), and the activity of some was widened to the G+ pathogen. Derivatives bearing shorter alkyl chains were potent against G−, while those with longer or multiple alkyl substituents were effective in G− as well as G+ strains. The level of hemolysis for all the derivatives was found to be 32 μg/mL or higher, which was greater than the MIC values of the most active QACs. Additional nephrotoxicity testing performed using human embryonic kidney cells HEK-293 revealed that toxicity toward mammalian cells is limited, and no measurable loss of cellular viability was detected up to the highest concentrations of 64 µg/mL. Finally, molecular dynamics simulations proved that the new agents preserved the ability to engage in specific interactions with LPS [66].

Vancomycin is a glycopeptide antibiotic that acts by disrupting cell wall synthesis. Boger and coworkers obtained vancomycin derivative **18a** alkylated at the *N*-terminus by nucleophilic substitution with methyl iodide (Figure 11). Its activity against *S. aureus* and enterococci was similar to or slightly higher than that of its parent drug (Table 2). Subsequently, (*p*-chlorobiphenyl)methyl substituent was introduced in a carbohydrate fragment in order to obtain compound **18b**, capable of inhibiting the bacterial transglycosylase enzyme. As anticipated, the modification resulted in a tremendous improvement in bioactivity, with MIC values from 0.15 to 1.2 µg/mL for *S. aureus* and *Enterococcus faecium*, respectively (Table 2) [67]. However, the novel vancomycin QACs were surprisingly inferior in comparison to previously reported *C*-terminus-modified QA vancomycin derivatives [103].

### 2.3. Carbohydrate Derivatives

Sugar derivatives also may exhibit antibacterial activity. For instance, rhamnolipid, which is naturally synthesized by *P. aeruginosa*, induces detachment of its biofilm and disperses biofilms produced by other species [104]. Peng et al. designed amphiphilic QA rhamnose **19** (Figure 12) to obtain agents that interfere with bacterial quorum sensing. Antibacterial and antibiofilm abilities were evaluated against two selected model bacteria, *P. aeruginosa* and *S. aureus*, by MIC identification, inhibition of biofilm formation, and destruction of the preformed biofilms. Recorded MIC values were moderate, with 39 and 625 µM for G+ and G− strain, respectively (Table 1). Moreover, this compound was effective against both bacterial biofilms through multiple mechanisms. It caused impairment of biofilm virulence as well as obtainment of antibiotic sensibility, which was evidenced by *P. aeruginosa* proteomic analysis [44].

Stevens and coworkers synthesized bolaamphiphilic derivatives starting from sophorolipids, compounds produced microbiologically by yeasts through fermentation of renewable resources. A set of QA biosurfactants **20** and **21** was obtained via reductive amination of sophorolipid aldehyde with diamines followed by quaternization with the use of alkyl halides (Figure 13). The highest activity was obtained for the peracetylated *N*,*N′*-dibutyl,-*N*,*N′*-dimethyl and *N*,*N*,*N′*,*N′*-tetrabutyl hexamethylene derivatives **20** (R^1^ = Me, R^2^ = *n*Bu, *n* = 5 and R^1^ = R^2^ = *n*Bu, *n* = 5). Their MIC values against methicillin-resistant *S. aureus* (MRSA) Mu50 strain (7.37 and 7.09 µM, respectively) [45] were better than for the reference antibiotics, vancomycin and clindamycin (6 and 1205 μM; respectively) [105]. The hybrid compounds showed weak activity against G+ strains *E. coli* and *K. pneumoniae* (MIC 227–472 μM). MIC values obtained for the latter compound are presented in Table 1 [46]. In the further step of the study, twelve new amphiphilic lipid-based QA compounds, **22** and **23**, bearing nitrogen atoms substituted with long alkyl groups (dodecyl, pentadecyl, and octadecyl) were obtained from oleic and petroselinic acid-based sophorolipids. Several of the derivatives displayed modest action against the G− bacteria *E. coli*, *P. aeruginosa*, and *Klebsiella pneumoniae*, with the highest activity for the deprotected QA petroselinic acid sophorolipids **23** (R = Me, *n* = 9, m = 9 or 12) with *N*-dodecyl and *N*-pentadecyl substituents, respectively (MIC 35.1–280 μM). MIC values of the latter compound are summarized in Table 1. All derivatives were moderately to highly active against G+ strains. The highest activity against G+ bacteria was shown by *N*,*N*-dimethyl-*N*-dodecyl petroselinic acid derivative **22** (R = Me, *n* = 9, m = 9), with extremely low MICs (Table 1). Increasing activities were obtained for the acetylated QA derivatives with a shortening of the chain length, while an opposite trend was observed for the deprotected QA sophorolipids. Unmodified oleic and petroselinic sophorolipid acids were weakly active and inactive against G+ and G− bacteria, respectively. The reported innovative compounds were proposed for application in the medical sector [46].

New QA macrolide analogs were obtained by Janas et al. via S*_N_*2 *N*-alkylation of clarithromycin desosamine nitrogen atom to introduce structurally diverse substituents (Figure 14). Overall, the modification was found to be beneficial and strongly enhanced water solubility (>2 mg/mL) at the expense of lower lipophilicity (clogP < 0) relative to the reference antibiotic, clarithromycin (~0.3 mg/mL; clogP 2.9). Antibacterial potency tests performed against a number of clinical and standard G+ bacteria including resistant strains revealed that compounds with small, less bulky, and relatively short unsaturated substituents such as allyl **24a**, crotyl **24b**, dimethylallyl **24c**, and alkyne **24e** were especially effective. The best antibacterial activity was reported for N-alkylammonium bromide **24e** against *Streptococcus pneumoniae* and *Streptococcus pyogenes* (MICs of 0.5 and 0.25 μg/mL, respectively; Table 2). Docking studies indicated favorable binding of QA clarithromycines in the ribosomal tunnel and showed that substituents attached to the quaternized nitrogen atom of the desosamine moiety enable stabilization of the π–π stacking interaction, which explains why compounds with longer and bulky substituents are less potent. Moreover, cytotoxicity assessed in a normal human dermal fibroblasts HDF cell line proved a more than 3.5-fold lower toxicity (IC_50_~70 μM) when compared to the reference drug [68].

### 2.4. Compounds Derived from Alkaloids and Natural Compounds

Bielawski et al. were inspired by the structures of QA alkaloids, such as chelerythrine sanguinarine, and berberine. They subjected amino alcohols to a double alkylation reaction with 1,2-bis(bromomethyl)benzenes and obtained a new class of N-spiro QACs, highly functionalized isoindolinio-tetrahydroisoquinolines (Figure 15). The compounds were evaluated against nine strains of bacteria, G+ cocci (*S. aureus*, *E. faecalis*, *Streptococcus mutans*, *Streptococcus salivarius*, *S. pyogenes*) and bacilli (*B. subtilis*), as well as some G− species (*E. coli*, *Moraxella catarrhalis*, *Campylobacter jejuni*). Interestingly, G− strains were more susceptible to the tested compounds than G+ bacteria. A majority of the synthesized derivatives showed the strongest antibacterial action, especially toward *M. catarrhalis* and *C. jejuni*; some of them were superior to norfloxacin against the latter strain. Minimum bactericidal concentration (MBC) to MIC ratio was above or equal to 4 for all the novel compounds. Based on the observation that QACs **25g**–**l** displayed higher antibacterial properties, it may be concluded that a phenyl substituent at the quaternary carbon stereocenter is favorable over cyclohexyl and methyl substituents. Moreover, the *p*-methoxyl moiety in the phenyl ring resulted in greater activity than in the case of *p*-bromo substitution. The influence of substituents in positions 3 and 4 of the phenyl ring at the isoindoline moiety was not significant in terms of antimicrobial action. Compound **25g** was found to be the most promising candidate for further studies since it presented slightly stronger antibacterial potential against *S. mutans* and *B. subtilis* than the control antibiotic (Table 2), although its activity was moderate. The assessed MIC values against most tested strains were in the range of 25–100 µg/mL, with the only exception of *Moraxella catarrhalis* (MIC of 10 µg/mL), and no activity was detected against *E. coli* up to 200 µg/mL. Nevertheless, a high level of selectivity was proved for the novel QACs as they did not possess cytotoxic, proapoptotic, or necrotic induction effects in normal cells, human skin fibroblasts CCD 1112Sk, up to the highest concentration tested of 200 µM, while the hemolytic activity was much above bactericidal concentrations [69].

Emodin is a natural compound that belongs to the anthraquinone family and is an active component that can be isolated from a traditional Chinese medicinal herb, rhubarb (Da Huang) [106]. Chalothorn et al. synthesized QA emodin analogue via structural modification of the aromatic ring—amination at the position 4 followed by double methylation (Figure 16). Antibacterial activity and cytotoxicity were evaluated against MRSA and noncancerous Vero cells, respectively. The novel QAC **26** was as potent as bare emodin against the tested bacteria (Table 2) in terms of bacteriostatic properties (MIC of 4 µg/mL) and presented a more effective bactericidal action (MBC of 64 µg/mL vs. >200 for the reference compound). What is more, the introduction of the QA group did not induce cytotoxicity against mammal cells—no inhibition in cell viability was detected at 50 µg/mL (IC_50_ of emodin was 42.5 µM) [70].

Vitamin B_6_ is engaged as a cofactor for many enzymes; therefore, pyridoxine derivatives might participate in numerous intracellular signaling pathways. Shtyrlin and coworkers designed and obtained a library of novel QA vitamin B_6_ derivatives **27** and **28** type, i.e., pyridoxines with six-membered acetals/ketals and a fragment of fatty acid (caproic, lauric, myristic, palmitic, or stearic acid) bound via a cleavable amide or ester linker (Figure 17). The antibacterial activity of the obtained QACs, especially towards G+ bacteria, is strongly correlated with their lipophilicity. This physicochemical feature was found to be essential for effective interaction with the hydrophobic bacterial membranes. Lead compounds **27** (R^1^ = propyl, R^2^ = H, R^3^ = C_11_H_23_, *n* = 2, X = NH) and **28** (*n* = 2, X = NH) exhibited antibacterial properties against laboratory and clinical G+ and G− bacteria comparable with that of BAC and higher than miramistin towards some selected strains (Table 2). Moreover, antibiofilm activity was confirmed by a drop in the colony-forming unit (CFU) number of biofilm-embedded bacteria. The mechanism of action comprises dose-dependent, fast depolarization of the bacterial membranes, as evidenced by membrane potential experiments. The compounds were investigated in terms of their genotoxicity and were found to be non-mutagenic in both the Ames test and the SOS-chromotest. Toxicity was evaluated in vitro against primary human skin fibroblasts HSF, human mesenchymal stem cells MSK, and HEK-293, and also in vivo on mice. The cytotoxicity of the most active derivatives, **27** and **28**, was relatively high (IC_50_ 0.15–3.21 µg/mL), although similar to BAC (IC_50_ 0.59–1.14 µg/mL). Acute oral (median lethal dose LD_50_ > 2000 mg/kg) and cutaneous (LD_50_ > 2500 mg/kg) administration studies revealed that in vivo toxicity was found to be quite low [71]. In the subsequent stage of the research, dihydropyrrole-containing acetals **29a**,**b** were obtained by alkylation of dioctylamine with pyridoxine dichlorides (Figure 17). The more lipophilic derivative **29b** demonstrated high antibacterial activity (Table 2) against the tested G+ bacteria (MIC of 0.5 μg/mL for *S. aureus*, *Staphylococcus epiderimidis* and *Micrococcus luteus*) including clinical MRSA isolates (MIC in the range of 2–16 μg/mL) and two G− strains (MIC of 2 and 0.5 μg/mL for *E. coli* and *Salmonella typhimurium*, respectively). Moreover, it was effective against G− clinical pathogens *Acinetobacter* spp., *Ralstoniae* spp., and *Klebsiella* spp. (MIC 1–32 μg/mL). Studies on the mechanism of action suggest that the possible antibacterial effect involves cell wall damage associated with the removal of Ca^2+^ ions from the membrane. The active compound did not show any DNA-damaging effect. However, it did not prove a selectivity profile either, as it was the most cytotoxic agent against HEK-293 cells (IC_50_ of 1.0 μg/mL) [72].

Similarly bis-ammonium pyridoxine derivative **30** was obtained (Figure 18) and evaluated biologically in terms of its antimicrobial efficacy. The logP values of the most active QACs were in the range of 1–3. Compound **30** (R^1^ = Et, R^2^ = H, R^3^ = C_12_H_25_), containing a dodecyl substituent on the QA nitrogen atom, was selected as a lead molecule as its antibacterial and antibiofilm activities were comparable to those of BAC (Table 2). The molecular target of this antibacterial agent was attributed to cellular membrane damage, similarly to other QACs. Despite relatively high in vitro cytotoxicity (HEK-293 IC_50_ of 2.81 µg/mL), in vivo effectiveness of 0.2% aqueous solution in the rat’s skin model was similar to the reference drugs, together with lower-than-BAC toxicity at oral administration on mice (LD_50_ of 1705 mg/kg) [73].

In the next stage of the study, pyridoxine-based QA derivatives of antimicrobial drug terbinafine were synthesized. The main idea was to use the pyridoxine moiety due to its properties that enhance transmembrane transport through mechanisms such as specific pyridoxine transporters in bacterial membranes [107] and to obtain molecules with easily adjustable physicochemical properties (i.e., steric volume and lipophilicity). Seven novel physiologically active conjugates were prepared by fusing the bromomethylpyridines with terbinafine into a single molecular construct via a QAC junction (Figure 19). The hybrids exhibited antimycotic and antibacterial activities against four fungal, three G+, and three G− bacterial strains (Table 2 and Table 4). The most interesting derivative, **33b**, showed inhibitory action against not only various bacteria and micromycetes in planktonic form but also microbial biofilm eradication properties comparable to conventional antifungals and antimicrobials. Moreover, the rate of spontaneous resistance development in four fungal and four bacterial strains treated by this compound was low. Assessment of the mechanism of action indicates that the compound exerts a bimodal effect, including targeting pyridoxal-dependent enzymes as well as damage to membrane integrity through a decline in the membrane potential and cell wall destruction associated with the removal of calcium ions. Unfortunately, the cytotoxicity assessed in human skin fibroblast cells revealed that the obtained hybrid (IC_50_ = 2.46 µg/mL), similar to other QACs such as miramistin (IC_50_ = 4.1 µg/mL) and benzalkonium chloride (IC_50_ = 2.1 µg/mL), is considerably toxic to eukaryotic cells, which limits the possibilities of its use in treatment [74].

Mikláš et al. reported the synthesis of new optically active amphiphilic QACs **34a**–**d** bearing a hydrophobic camphor moiety (Figure 20). These homochiral QACs were synthesized starting from (1*R*,3*S*)-(+)-camphoric acid and presented broad-spectrum antimicrobial activities, not only antibacterial but also antifungal. They were tested against microorganisms such as *S. aureus*, *E. coli*, and *C. albicans*. QACs **34a**–**c** were found to be stronger antimicrobial agents compared to clinically used benzalkonium bromide. The most preferable activity was noticed for compound **34c** (Table 1 and Table 3) with 16 carbon atoms in the alkyl chain [47].

Quinuclidine is a bioactive bicyclic saturated alkaloid precursor. Sprung and coworkers synthesized quaternary ammonium 3-hydroxyquinuclidinium salts **35a**–**g** with various lengths of alkyl chain ranging from 3 to 14 carbon atoms by the quaternization reaction of quinuclidine-3-ol and appropriate halogenoalkanes (Figure 21). The antimicrobial potential of the novel QACs was surveyed towards a series of 16 strains of pathogenic bacteria, yeast, and molds, including clinical multidrug-resistant ESKAPE isolates as well as emerging food spoilage. The results showed that the addition of at least a 10-carbon chain is critical to ensure antimicrobial activity. QAC **35g** bearing a 14-carbon alkyl substituent was identified as the most efficient agent, highly active against a wide spectrum of all pathogens tested. MICs values determined for Gram-positive bacteria ranged from 0.06 to 3.9 µg/mL, while values for fungal strains were between 0.12 and 3.9 µg/mL (Table 2 and Table 4). This compound targeted the cell membrane, causing disruption of the structures and leading to lysis, as was shown by the atomic force microscopy images, flow cytometry, and fluorescence microscopy. Longer treatment time resulted in complete destruction of the cell population, while at concentrations higher than MIC, bacteria were killed instantaneously. The cytotoxic effect of the most active QAS on noncancerous epithelial cells was moderate [75].

Burilova et al. prepared quaternary derivatives of quinuclidine in the same manner; however, the synthesized QACs were more lipophilic with tails of 14, 16, and 18 carbon atoms (Figure 1). The antimicrobial activity of the obtained QACs was evaluated against Gram-positive (i.e., *S. aureus* and *Bacillus cereus*) and negative bacteria (*P. aeruginosa* and *E. coli*) as well as fungi (*A. niger*, *Trichophyton mentagrophytes* var. *gypseum* and *C. albicans*). The derivatives displayed interesting antibacterial and antifungal activity, especially towards Gram-positive strains. QAC **36b** not only exhibited the highest bactericidal activity (Table 2) among the investigated compounds but also demonstrated 6- and 15-fold greater potency than the known antibacterial agent, norfloxacin, against *S. aureus* and *B. cereus*, respectively. Antimicrobial activity of QAC **36c** was similar to two reference drugs, norfloxacin and ketoconazole. The latter drug showed a minimum fungicidal concentration of 4 μg/mL against the tested fungal strains, while the values of compound **36c** were in the range of 2–4 μg/mL (Table 4). Derivative **36b** was tested against the human immortalized epidermal HaCaT and HEKa skin cell line up to the concentration of 8 μg/mL, and cell viability at the highest concentration tested only decreased by approximately 10–15%, indicating its safety for use on the skin. The proposed cationic surfactant systems based on quinuclidine were proposed as multifunctional biocompatible compounds with potential in biotechnology and nanomedicine [76].

### 2.5. DABCO Derivatives

Bogdanov and coworkers developed a method for the synthesis of antimicrobial QA acylhydrazones [77,108,109,110,111] containing the quinuclidine analogue, i.e., 1,4-diazabicyclo[2.2.2]octane (DABCO). QACs **37a**,**b** were synthesized in the condensation reactions of *N*-substituted isatin with hydrazides (Figure 22) in good to excellent yields (73–96%). The compounds demonstrated efficacy against certain Gram-positive strains, with the DABCO derivative **37a** proving to be the most potent. It displayed a four-fold increase in efficiency compared to chloramphenicol, with a MBC of 15.6 µg/mL against *S. aureus*. Moreover, the described QACs showed no hemolytic effects [77].

Wuest and colleagues utilized a cost-effective approach outlined in Figure 23 to synthesize QA DABCO derivatives **38**–**40**, featuring diverse lipophilic substituents. These compounds demonstrated noteworthy micro- and sub-micromolar efficacy against a spectrum of pathogens, encompassing Gram-positive bacteria such as *E. faecalis* and *S. aureus* (including MRSA strains) as well as Gram-negative bacteria like *E. coli* and *P. aeruginosa*. Among the tested compounds, QA DABCO derivative **38** emerged as the most potent (as detailed in Table 1) in this investigation, displaying selectivity. Notably, its therapeutic index, calculated as the ratio of red blood cell lysis concentration (<20%) to MIC against MRSA, was determined to be 4 [48].

Cardanol is a phenolic compound obtained as a by-product from the cashew nut processing industry that can be easily functionalized. Wang and colleagues utilized this renewable biomass raw material to synthesize novel QA derivatives **41** and **42a**,**b**, presenting them as promising alternatives to existing antiseptics and cationic surfactants derived from fossil fuels (Figure 24). DABCO derivative **42b** proved to be more active than its less lipophilic analogue **42a**. However, due to poor water solubility this molecule was not the most active congener within the series. Hence, morpholinium derivatives **41** and **43** proved to be not only highly soluble but also effective in eradicating *S. aureus*, *B. subtilis*, and *E. coli* bacteria at concentrations of 32 µg/mL or lower. Among them, the most potent QAC **43** demonstrated MICs of 32, 8, and 16 µg/mL against *S. aureus*, *B. subtilis*, and *E. coli*, respectively (Table 2). Scanning electron microscopy (SEM) images clearly depicted disorganized and damaged pathogenic bacteria cells with irregular shapes. The findings suggest that these compounds can interact with the cell membrane, leading to structural disintegration and the formation of pores, ultimately causing the escape of cytoplasmic components such as autolysins and resulting in membranes dissolution. Consequently, these destructive effects culminate in bacterial cell death [78].

Pashirova et al. synthesized double DABCO QACs **44** with two hydrophobic chains and various alkyl linkers (Figure 2). Antimicrobial studies showed that the activity of the tested surfactants was strongly affected by their structure and improved with decreasing alkyl chain length from octadecyl to dodecyl analogue. Compound **44** with 2-carbon linker (*n* = 1, R = C_12_H_35_) was the most effective compound in terms of bacteriostatic and bactericidal action (MIC = 15.6, 0.98, and 0.48 μg/mL against *E. coli*, *B. cereus*, and *S. aureus*, respectively), especially towards G+ bacteria (Table 2), without hemolytic effect at a concentration of 3.1 μg/mL. The compound was proposed for the preparation of cationic liposomes containing rhodamine B, intended for transdermal drug delivery [79].

### 2.6. Gemini Surfactants and Polycationic QACs

Gemini surfactants (GS) are composed of two symmetric hydrophilic moieties (polar cationic ammonium heads) covalently joined by a spacer and a hydrophobic part that is usually constituted by alkyl tails of different chain lengths. Xu et al. synthesized ester-based GS **45** through a straightforward acylation reaction (Figure 25). Highly lipophilic GS **45d**,**e** bearing 16–18 carbon atom tails were more active against a G+ *S. aureus* strain (MIC < 2 µg/mL), while less hydrophobic C10–C14 derivatives **45a**–**c** exhibited better activity against G− *E. coli* bacteria (MIC of 32 µg/mL) (Table 2) [80]. In turn, Liu and collaborators obtained amide-linked GS **46** bearing diverse counterions (Figure 25). Bacteriostatic activity against *P. aeruginosa*, *E. coli*, and *B. subtilis* was higher at 50 µg/mL than that of BAC, even at 100 µg/mL [112,113,114].

Akram and coworkers designed and synthesized eco-friendly ester-ether GS **47** (Figure 26). Due to the presence of a cleavable spacer, the obtained GS was found to be biocompatible and biodegradable. The antimicrobial properties of the novel GS were verified using the agar well diffusion method. Moreover, their hemolytic activity was low, as evidenced by determined half-maximal hemolytic activity (HC_50_) in the range of 223–315 µg/mL [115].

Taleb et al. synthesized eco-friendly surfactants featuring 4-alkyloxybenzene moieties connected to ammonium headgroups through a biodegradable amide linkage. The applied synthetic pathway involved the efficient quaternization of *N*,*N*,*N*′,*N*′-tetramethylalkylenediamine, serving as a hydrophobic linker, with bromoacetanilides. (Figure 27). Antimicrobial results showed that high water solubility and optimal hydrophobic/hydrophilic balance of the molecule were the key parameters enabling interactions with the bacterial membrane surface. Inhibition of *S. aureus* growth was promoted with elongation of the hydrophobic chain from 8 to 12 carbon atoms, with the maximal antibacterial activity achieved for derivative **48** bearing an alkyl linker with 4 carbon atoms (*n* = 12, m = 4; MIC of 17 μM). In contrast, *E. coli* growth was more efficiently suppressed by less lipophilic compound **48** (*n* = 10, m = 2; MIC of 74.7 μM) (Table 1) [49].

The diverse molecular mechanisms through which *N*-chloramines exhibit antimicrobial activity include protein targeting. Specifically, they interfere with hydrogen bonds, alter the structure-governed functions of proteins, and attack amino acids containing sulfur, such as methionine and cysteine. Additionally, *N*-chloramines can penetrate bacterial cells, causing damage to cell wall proteins and intracellular vital components [116,117]. Compounds of this nature can be utilized in the synthesis of antibacterial QACs. For instance, Liu and colleagues successfully produced composite biocides incorporating QAC/*N*-chloramine, demonstrating fast killing kinetics against MRSA, *E. coli*, and *P. aeruginosa* [118]. Li et al. synthesized novel *N*-chloramine GSs **49a**–**d** by *N*-alkylation of 5,5-dimethylhydantoin followed by quaternization of tetramethylethylenediamine and chlorination of the formed mid-products (Figure 28). The synthesized GSs, featuring diverse alkyl linkers, exhibited effective antimicrobial properties against both *S. aureus* and *E. coli*. Among the active analogs, **49d**, with a 10-methylene linker, demonstrated the highest biocidal efficacy [119]. Due to a complete elimination of *S. aureus* and *E. coli* CFU within 5 min of contact at a concentration of 20 µg/mL, this derivative demonstrated superiority over previously reported zwitterionic *N*-chlorohydantoins [120]. Additionally, its effectiveness was comparable to the mono QAC obtained in the previous studies [121].

Wuest’s research team obtained dicationic derivatives through hybridization of BAC and cetylpyridinium chloride (CPC) moieties [50] or through quaternization of 1,1′-bis(dimethylaminomethyl)ferrocene [51] (Figure 29). The synthesized antibacterial compounds **50** and **51** demonstrated consistent efficacy across a spectrum of both G+ and G− strains. Generally, resistance to MRSA was not detected. Among the compounds tested, **50d** and **51c** emerged as the most potent in BAC-CPC hybrids and ferrocene sets, respectively, as highlighted in Table 1. In contrast, dicationic pyrrolidine-based bolaamphiphiles **52a**–**c** also obtained in the QAC study demonstrated poor antibacterial activity (MICs > 250 µM towards all tested bacteria) [52].

Rohand et al. prepared QA bis-1,3,4-oxadiazoles **53a**,**b** via a multi-step procedure (Figure 30). The compounds were evaluated against a panel of G+ and G− bacteria including MRSA pathogens. Antibacterial activity against some microorganisms was high, especially towards G− *Citrobacter freundii* (MIC of 0.27 and 0.10 µg/mL for **53a** and **b**, respectively) and MRSA (MIC of 0.98 µg/mL). On the contrary, the activity was notably weaker against *E. coli* (MIC of 100 and 50 µg/mL for **53a** and **b**, respectively) (Table 2) [81].

Multifunctional QACs **54** with tunable hydrophilic/hydrophobic balance were designed by Dey et al. (Figure 31). The compounds incorporated two QA groups, ethanol moieties, nonpeptidic amide bonds, pendant alkyl chains, and a lipophilic aliphatic linker. The length of external chains was found to have a dominant impact on antibacterial activity, while the effect of the spacer arm was lower. Compounds bearing 10-carbon atom chains and 3- to 10-methylene linkers were in general the most active (MICs of 0.5–1, 1–2, 1–2, 2–4, and 2–8 µg/mL against MRSA, *E. coli*, *Acinetobacter baumanii*, *K. pneumoniae*, and *P. aeruginosa*, respectively). However, these compounds exhibited undesirable hemolytic activity, with HC_50_ values in the range of 18–34 µg/mL. For instance, dodecyl derivative **54** (m = 12, *n* = 8) exhibited remarkable potency (Table 2); however, its hemolytic activity was elevated, reaching 12 µg/mL. Conversely, QAC **54**, with dodecyl substituents and hexane linker (m = 12, *n* = 6), was found to be highly potent against MRSA and *E. coli* (MIC in the range of 1–2 µg/mL), showing maintained selectivity against bacterial cells with HC_50_ of 577 µg/mL. Moreover, its IC_50_ values towards mammalian Madin–Darby canine kidney MDCK epithelial and A549 cells were 188 and 194 µg/mL, respectively. This antibacterial agent **54** was tested against an extended panel of MRSA and vancomycin-resistant *S. aureus* (VRSA) strains showing MIC values in the range of 1–2 µg/mL. It not only showed bactericidal activity against planktonic MRSA cells but also eradicated its biofilms. Membrane permeabilization and depolarization studies revealed that its molecular target is the bacterial membrane, while plasma and serum stability tests proved that it is less susceptible to degradation than AMP under physiological conditions [82].

Wan et al. designed and synthesized fluorescent stilbene QACs **55a**–**f** with large space steric hindrance from *N*,*N*-dimethyldodecylamine, 4,4′-diaminostilbene-2,2′-disulfphonic acid, and 2,4,6-trichloro-1,3,5-triazine (Figure 32) as antibacterial and whitening agents. The novel QACs were fairly stable in aqueous solutions and exhibited an efficient optical whitening effect as well as significant antibacterial activity. Among them, compound **55d** turned out to be the most potent molecule against the selected strains, with MICs of 16 µg/mL for *E. coli* and 4 µg/mL for *C. albicans* (Table 2 and Table 4) [83,122].

Dicationic organosilanes **56** were synthesized by Ahn and collaborators as surface-modifying disinfectants (Figure 3). The utilized compounds were found to form stable antibacterial coatings on various surfaces and were active towards both G+ and G− bacteria. The effectiveness of novel compounds was comparable to traditional antiseptics; therefore, they can be used as immediate surface disinfection agents with durable and long-lasting antimicrobial potency [123].

Xu and coworkers proposed compounds that were applied as antibacterial softening agents for paper processing. They synthesized alkyl QACs **57a**–**c** bearing three QA groups in a two-step process. Thus, triethanolamine was first reacted with sulfuryl chloride, and subsequently, the formed intermediate underwent quaternization with tertiary amines featuring various long alkyl chains (Figure 33). Inhibition zone experiments indicated that the obtained materials exhibit antibacterial properties against *S. aureus* and *E. coli*. Therefore, the authors suggest that these materials could have potential applications in antibacterial tissues or daily disposable sanitary products to prevent microbial contamination [124].

Bazan and collaborators synthesized cationic conjugated oligoelectrolytes **58** and **59** with four QA moieties (Figure 34). The designed molecules maintained a fixed distyrylbenzene framework but varied in the length of substituents attached to the cationic site and the spacer between the core and the QA group. Antimicrobial efficacy determination against *E. coli* revealed that compound **58**, bearing four or more carbon atoms in the terminal alkyl chain, is a potent antibacterial with an MIC of 4 µg/mL. Among the most active derivatives, compound **58**, possessing a 4-methylene linker (*n* = 4, m = 4), was found to be the least cytotoxic against mammalian HepG2 cells with IC_50_ value above 1024 µg/mL. The compound with an optimal structure exhibited negligible hemolysis of red blood cells below 5% at 1024 µg/mL and bactericidal efficacy in time-kill assays [84]. In this subsequent investigation, amide derivatives **59** were synthesized and subjected to biological evaluation (Figure 34). Propyl analogue **59b** was found to be effective against *E. coli* (MIC = 2 µg/mL, Table 2) and stable in the presence of human serum albumin. This compound presented low toxicity (IC_50_ of 740 µg/mL against HepG2) and neglectable hemolytic activity (HC_50_ > 1024 µg/mL). The molecular mechanism of bactericidal action was associated with outer membrane permeabilization and cytoplasmic membrane depolarization [85].

Pisárčik and coworkers synthesized gemini QACs with linear alkyl chains and variable polyethylene spacers. The compounds presented in Figure 4 were obtained in the reaction of α,ω-dibromoalkanes with *N*-tridecane-2-yl-*N*,*N*-dimethylamine. The compounds **60e**,**f** bearing 8 and 10 methylene spacers showed microbiocidal activity against *E. coli*, *S. aureus*, and *C. albicans*, superior to the reference compounds, cetylpyridinium bromide, benzyldodecyldimethylammonium bromide, and carbodependecinium bromide. Compound **60d** was the most effective against the tested bacterial strains (Table 1), while congener **60e** was able to sufficiently eradicate the evaluated fungal microorganisms (Table 3) [53].

Zhou et al. obtained gemini QACs **61a**–**d** from methyl esters of fatty acids (dodecanoic, hexadecanoic, tetradecanoic, and octadecanoic), triethylenetetramine, and ethyl bromide (Figure 35). The efficacy of the four novel derivatives was assessed against both G+ (*B. subtilis*, *S. aureus*) and G− bacteria (*E. coli*), as well as fungi (*Aspergillus flavus* and *C. albicans*). The synthesized surfactants were found to exhibit corrosion inhibition efficiency and biocidal properties that increase with the elongation of the alkyl chain. The authors suggest that the proposed QACs could find potential applications in the oilfield industry [125].

A series of water-soluble dimeric quaternary ammonium surfactants **62a**–**h** were prepared by Brycki and coworkers. Hence, the bromomethylbenzene was reacted with tertiary alkyldimethylamines via the S_N_2 mechanism to give QACs in good yields (Figure 36). The undertaken systematic study involved structure and surface properties evaluation as well as antimicrobial properties examination. MIC values determined against bacteria *B. subtilis*, *S. aureus*, *P. aeruginosa*, and *E. coli* as well as microscopic fungi, i.e., *A. niger*, *C. albicans*, and *Penicillium chrysogenum* were in the range of 12.2 to 12,500 µM. The observed biocidal effectiveness depended on the length of the alkyl groups. Derivatives bearing shorter hydrocarbon chains (**62a**,**b**) were not able to penetrate the bacterial cells (MICs from 6250 to above 12,500 µM). The strongest activity antibacterial was observed for QACs with 10–12 carbon atoms (**62d**,**e**) in the alkyl substituent, while the antifungal properties were the highest in the case of compounds that contained 10–14 (**62d**–**f**) carbon atom chains. The extension of a lipophilic substituent beyond 14 methylene groups led to a reduction in antimicrobial activity. Decyl and dodecyl derivatives **62d** and **e** were found to exhibit the highest antibacterial and antifungal potencies, respectively (Table 1 and Table 3) [54].

The research team led by Brycki employed also a similar methodology to procure gemini QACs **63a**–**h** (Figure 36), which were tested against *E. coli* and *S. aureus* bacteria. *C. albicans* yeast, as well as *A. niger* and *P. chrysogenum* molds. QACs with the shortest alkyl chain (**63a**) displayed weak activity as evidenced by the highest values of MIC (>3750 µM for all the strains tested). In contrast, compound **63e**, which features 12 carbon atoms in the alkyl substituents, proved to be the most active microbiocide in the series and superior to antibacterial GS bearing aromatic phenyl spacer **62a**–**h** (Table 1 and Table 3). Further elongation of alkyl chains led to decreased activity, revealing a distinct cut-off effect characterized by a parabolic correlation between compound potency and lipophilic character [126].

Kowalczyk et al. synthesized gemini surfactants incorporating azapolymethylene spacer **64a**–**u** in the alkylation reactions of tertiary diamine with halogenoalkanes (Figure 37). The MIC values of the synthesized compounds were determined against bacteria (*S. aureus*, *P. aeruginosa*, *E. coli*, and *B. subtilis*) as well as fungi (*A. niger*, *C. albicans*, *P. chrysogenum*). Similarly, QACs with the shortest alkyl substituents displayed the weakest activity. The most pronounced antimicrobial properties were exhibited by dodecyl and tetradecyl derivatives. Compound **64l** was found to be the most promising antibacterial agent, while derivative **64t** was the most effective against fungal strains (Table 1 and Table 3). It is worth emphasizing that the compounds exhibiting the highest activity both feature substituents with a 12-carbon atom chain [55].

Sikora and coworkers obtained gemini QACs derived from 1,4:3,6-dianhydro-D-mannitol **65a**–**d**. These environmentally friendly molecules can be gained in large quantities from natural sources as a byproduct of the agriculture industry. Their objective was to obtain biodegradable, nontoxic compounds of high biological activity. The compounds were prepared via a two-step synthetic route. Firstly, 1,4:3,6-dianhydro-D-mannitol was converted into di-*O*-trifluoromethanesulfonyl intermediate and subsequently subjected to the reaction with a variety of tertiary aliphatic amines (Figure 38). The newly synthesized compounds displayed relatively weak antimicrobial properties which increased with the elongation of the carbon chain length. The majority of the QACs exhibited MIC values above 64 and 1024 µg/mL for bacterial and fungal strains, respectively. The sole exception was the moderately active decyl analogue **65d**, with MICs ranging from 4 to 32 µg/mL (Table 1, Table 2, Table 3 and Table 4). However, considering its toxicity (IC_50_ of 12.8 µM against HaCaT keratinocytes), this compound cannot be regarded as a selective agent. None of the QACs were mutagenic in the Ames test up to the highest concentration tested [56].

### 2.7. Methacrylate Monomers for Dental Applications

QACs can be employed in dentistry to reduce the biodegradation of dental composites induced by oral microorganisms. This approach enhances the durability of restorations and hinders the occurrence of secondary caries at the restoration margins. This is achieved by reducing bacterial adhesion and inhibiting biofilm formation on the surface of the resin composite. A composite matrix is usually composed of methacrylate monomers that can be polymerized under UV light. Sun and colleagues synthesized innovative antimicrobial methacrylates containing QA groups (Figure 39). Their efficacy against both G+ and G− bacteria, as well as fungi, was demonstrated, along with their capability to undergo polymerization. The dodecyl derivative **66d** exhibited the highest activity against the tested bacteria and fungi, with minimum inhibitory concentrations (MICs) ranging from 0.8 to 25 µM (Table 1 and Table 3). However, upon polymerization, the macromolecular product derived from derivative **66a** demonstrated the most significant antimicrobial activity within the series, with MICs ranging from 100 to 400 µM against the tested strains [57]. Another research group, led by Cherchali, synthesized a monomer using the same scaffold, dimethyl-hexadecyl-methacryloxyethyl-ammonium iodide **66f**, and integrated it as an antibacterial monomer into the experimental methacrylate-based dental composite matrix. The addition of 7.5% **66f** into the composite material resulted in a significant antibacterial effect against cariogenic bacteria responsible for tooth decay, such as *S. mutans*. This resulted in a significant reduction of approximately 98% in CFU and a nearly 50% decrease in metabolic activity, coupled with the inhibition of biofilm formation. Additionally, the mechanical properties of the composite were found to be satisfactory [99].

Xu and collaborators designed a new antibacterial methacrylate monomer **67** containing a QA group, an aromatic substituent, and a long-chain aliphatic linker. The formed bromide counter ion was replaced in order to obtain fluoride-releasing dental material with a fluoride ion source (Figure 40). In addition, this modification enhances color stability by preventing the formation of oxidized colored molecules, such as Br_2_ or I_2_. The cytotoxicity tests performed with the application of L-929 mouse fibroblasts revealed that the obtained compound is biocompatible. The synthesized monomer showed a bactericidal effect against *S. mutans* and *Lactobacillus casei*. The composite containing 3% of this compound exhibited significant antibiofilm activity, leading to a drastic reduction in the amount of biofilm formation by three orders of magnitude (killing rate exceeding 99.9%), with no significant adverse effects on its mechanical and physical properties [127]. In the subsequent stage of the study, cross-linking dimethacrylate monomers **68a**–**c**, capable of forming three-dimensional structures that stabilize the polymeric network, were synthesized (Figure 40) and evaluated for their efficacy against bacterial pathogens. Monomers **68b**,**c** generally demonstrated high antibacterial activity, with the best efficacy shown for more lipophilic compound **68c** containing a long hexadecyl carbon chain. Both monomers were strongly active towards *S. mutans*, *L. casei*, and *S. aureus* at a concentration of 10 µM and effective towards *P. aeruginosa* at 100 µM. Biocompatibility studies against human gingival fibroblasts indicated that the compounds are safe at 10 µM and cause about 20 and 60% decreases in cell viability at 10 µM, respectively. The proposed QACs are expected to enhance the anticaries efficacy of dental composites and prolong their service life [128].

Manouchehri et al. also synthesized dimethacrylate dental monomers; however, the proposed structures **69a**,**b** contained bis-QA moiety (Figure 41). The obtained GS exhibited MIC values against *S. mutans* of 6.25 and 3.12 µg/mL, respectively. The addition of the monomers at a concentration of 1% to the commercial adhesive did not significantly affect the bonding properties of the carrier material or adversely influence the degree of monomer-to-polymer conversion. Nonetheless, cytotoxicity evaluation against human foreskin fibroblasts (HFF2) revealed some reduction in cell viability [86].

A similar approach was undertaken by Cadenaro and coworkers. They synthesized nine novel di-methacrylate monomers based on bis-QACs **70a**–**h** (Figure 5) and evaluated their antimicrobial properties against both G+ (*S. mutans*, *S. aureus*, *Streptococcus mitis*, and *Streptococcus sanguinis*) and G− (*E. coli*) bacterial strains. Overall, achieving a balance between hydrophilicity and hydrophobicity was crucial in designing effective biocides, as it directly influenced the antibacterial range of action. Moreover, a flexible twelve-methylene spacer between two QA groups proved advantageous for biocidal activity compared to the rigid aromatic ones found in **70g**,**h**. The introduction of the phenyl moiety enhanced the lipophilicity of the **70f** monomer, resulting in the highest antibacterial activity. The MIC and MBC values for the tested microorganisms ranged from 5 to 20 μg/mL. Moreover, this compound effectively inhibited *S. mutans* biofilm at a concentration of 5 μg/mL, equivalent to the MIC value. The QACs demonstrated low cytotoxicity, remaining well-tolerated up to 50 μg/mL on the human dental pulp stem cell model, affirming their excellent biocompatibility within the oral environment. In summary, the suggested monomers exhibit promising potential for application in dental materials, potentially preventing bacterial proliferation at restoration margins [87].

### 2.8. Other Antibacterials

Nitric oxide (NO) plays a crucial role as a signaling molecule in regulating biofilm formation. It induces a transition to the planktonic mode of growth and prevents the initial cell aggregations [129]. There are many molecules that may release free NO, causing bacterial viability reduction, and benzofuroxans are one such example. Chugunova and her collaborators synthesized QA salts **71a**–**d** as a mixture of two isomers during the reaction process, which involves the synthesis of a tertiary amine benzofuroxan derivative and its subsequent reaction with benzyl bromide or dibromoxylenes (Figure 42). Only one bromomethyl group of *o*-, *m*-, and *p*-dibromoxylenes participated in the quaternization reaction. None of the novel compounds showed toxicity, as hemolysis at the minimum inhibitory concentration (MIC) did not exceed 1%. The isomeric mixture **71d** displayed the highest biological activity, with MIC values two to four times lower than the reference drug chloramphenicol, exclusively against G+ strains, i.e., *S. aureus* and *B. subtilis*. However, these derivatives exhibited inactivity towards G− and fungal strains (Table 2) [88].

Photochemical NO generators are particularly appealing due their ability to precisely release NO in a controlled, light-triggered fashion with accurate spatiotemporal control. Consoli and coworkers designed and synthesized a novel multivalent photoresponsive molecular construct, **72**, by circularly grafting multiple units of a NO photodonor, 3-(trifluoromethyl)-4-nitrobenzenamine, onto the upper rim of a calix[4]arene backbone. Subsequently, they added two quaternary ammonium groups at the lower rim of the polyphenolic macrocycle by covalent linkage to obtain a clustered and spatially organized conjugate **72** (Figure 43). QA moiety was introduced to confer water-solubility, while *N*-dimethylethanol portion was chosen as the cationic polar head group for its well-known ability to penetrate bacterial lipid membranes without harmful effects on eukaryotic cells [130]. The obtained compound was soluble in hydroalcoholic solutions and formed nanoaggregates able to efficiently generate NO under the exclusive switch of visible irradiation inputs. Free NO molecule formation was strictly light-controlled—it started in the presence of irradiation, stopped immediately in the dark, and started again once the light was switched on. The nanoconstruct demonstrated its antibacterial efficacy under light stimulation, as evidenced by a significant reduction in bacterial load for both *S. aureus* and *E. coli*, representing Gram-positive (G+) and Gram-negative (G−) strains, respectively. A reduction in G+ bacteria was observed in the dark after 30 min, while the biocidal effect became more pronounced with visible light irradiation after 10 min, ultimately achieving a nearly 100% decrease in colony-forming units (CFU) after 20 min. The compound exhibited an effective decrease in *E. coli* bacteria after 30 min of hydro-alcoholic solution irradiation, with no significant changes recorded in the dark. Control experiments conducted with the mammalian dermal cell line HSF showed insignificant anti-proliferative activity (<15%) during up to 30 min of irradiation. The novel hybrid molecule presents an appealing alternative to traditional antibacterials in combating the phenomenon of antimicrobial resistance [131].

Padnya et al. utilized the calixarene skeleton to synthesize novel antibacterial QA constructs **73a**–**e**. They employed a thiacalixarene macrocyclic platform in a two-step procedure to prepare a series of tetrasubstituted amidopropylammonium QACs with substituents of varying lengths and natures (Figure 44). All the in situ compounds obtained based on *p-tert*-butylthiacalix[4]arene were converted into chloride analogs using ion-exchange resin to eliminate the influence of anions on bioactivity. The newly derived compounds exhibited inhibition of growth against the studied G+ bacteria, comparable to known commercial antiseptics such as BAC, miramistin, and antiseptic chlorhexidine (Table 2). Among the most active compounds, derivative **73b** showed effectiveness against *Staphylococcus* bacteria when additionally tested against clinical isolates. The QACs evaluated against human skin fibroblast (HSF) cells demonstrated lower toxicity compared to reference drugs. The most promising antibacterial agent exhibited the highest selectivity index (SI) values due to its lack of cytotoxicity (IC_50_ above 1000 µg/mL). Studies involving a lipid membrane model indicated that compounds in the 1,3-alternate conformation, bearing a lipophilic alkyl fragment, were incorporated into the membrane, leading to the “clumping” of liposomal vesicles [89].

Hybrid drug development is one of the strategies employed to combat antimicrobial resistance. A hybrid drug molecule consists of diverse bioactive fragments fused into one compound. The various pharmacophoric parts act against different molecular targets; consequently, there is a reduced risk that bacteria will mutate and evolve defense mechanisms against multiple drug units. Remarkably, fluoroquinolones fused with other antibacterials constitute the most comprehensively described hybrid compounds [132]. Fedorowicz et al. synthesized QA quinolones **74a**–**h** and **75a**–**h** by means of tandem Mannich-electrophilic amination reactions from profluorophoric isoxazolones [133] and antibiotics bearing a secondary amine moiety at position 7 of the quinoline ring (Figure 45). The obtained zwitterionic hybrids were noticeably more hydrophilic than the parent fluoroquinolones [134,135]. The obtained QACs were screened in vitro for their antimicrobial, bactericidal, and antibiofilm activities against a panel of G+ and G− bacterial pathogens. Ciprofloxacin (**75e**) and lomefloxacin (**75d**) conjugates displayed extremely low molar-based antibacterial potency comparable to the unmodified drugs (Table 2) and considerable activity in *E. coli* DNA gyrase inhibition experiments with IC_50_ values in the low micromolar range. Moreover, the derivatives demonstrated a low propensity for bacterial resistance development and did not show notable cytotoxicity trends toward the HEK-293 control cell line. Finally, a molecular docking study revealed that the synthesized QACs could bind to the active sites of bacterial topoisomerase IV and DNA gyrase in a manner characteristic of fluoroquinolones [90].

In the extended study, dimethyl QA fluoroqionolone derivatives **76a**–**h** were synthesized (Figure 46). The obtained compounds exhibited antibacterial and antibiofilm potencies against *S. aureus* and *P. aruginosa*, with the most active ciprofloxacin derivatives being **76e**,**g** (MICs of 6.25 μM, Table 1). Compound **76e** was additionally effective against biofilms of both pathogens in pre-exposure as well as post-exposure modes. Its activity was superior to ciprofloxacin in reducing the number of viable cells in mature biofilms. The compounds were not toxic against mouse embryonic fibroblasts BALB/3T3 clone A31 up to 200 μM, providing a selectivity index above 32 and confirming their safety in mammals [58].

Wang and colleagues synthesized QACs **77** by integrating a nitrogen-containing heterocycle, 5-phenyl-1,3,4-oxadiazole-2-thiol, along with short, medium, or long alkyl linkers such as ethyl, propyl, butyl, hexyl, octyl, decyl, or dodecyl (Figure 47). The antibacterial, antifungal, and cytotoxic activities, as well as the mechanism of action, were experimentally evaluated. The most potent compounds in each series were the lipophilic derivatives bearing the dodecyl chain (*n* = 8). They displayed fairly good antimicrobial effects against common pathogens such as *S. aureus*, *a-H-tococcus*, *E. coli*, *P. aeruginosa*, *Proteus vulgaris*, and *C. albicans*, especially **77a** (R = H, R^1^ = R^2^ = Et, R^3^ = CH_2_CH_2_OH, *n* = 8) and **77b** (R = *t*Bu, R^1^ = Me; R^2^ = R^3^ = CH_2_CH_2_OH, *n* = 8) (Table 2 and Table 4). It should be noted that the activities of **77b** were nearly identical to those of the positive control (BAC), with MBC values equal to corresponding MICs. Cytotoxicity experiments conducted in HaCat and human normal liver LO2 cell lines indicated that cytotoxicity and antimicrobial properties increased concomitantly with the chain length from hexyl to the dodecyl linker. Nevertheless, all the compounds exhibited lower cytotoxicity compared to BAC. Additionally, most of the compounds were found to be safe, causing no hemolysis at concentrations below 50 µg/mL. Treatment of *E. coli* and *S. aureus* bacteria led to morphological and physical alterations, including cell wall surface deterioration, separation of cell wall layers, wrinkled abnormalities, detrimental effects on the cell membrane, and significant surface collapse. SEM and TEM images revealed leakage of intracellular substances, cytoplasm release, and numerous incomplete cells. Furthermore, a decrease in electron density was observed in both the cell envelope and cytoplasm, accompanied by the presence of large quantities of electron-dense extracellular materials. The antibacterial mechanism was attributed to QAS adhesion and fixation on cell walls and membrane surfaces, leading to puncturing and disruption of peptidoglycan formation, ultimately resulting in the release of bacterial cytoplasm. Structure–activity relationships suggested that the long chain and two flexible hydroxyethyl groups in QAS facilitated effective passage into the cell through the membranes, enzyme inactivation, and bacterial damage [91,92].

Tuo et al. synthesized 1,3,4-oxadiazole-morpholinium QACs (**78**), demonstrating their potential as effective antibacterial agents in agriculture, specifically for combating plant diseases (Figure 48). The designed molecules exhibited good to excellent antibacterial activity, with particular emphasis on the dodecyl compounds containing 3-chloro and 4-methyl substituents **78** (R = C_6_H_4_-3-Cl or C_6_H_4_-4-Me, *n* = 12). Bioassay results revealed the antibacterial efficacy of these compounds with IC_50_ values ranging from 1.40 to 2.80 μg/mL against *Xanthomonas oryzae* pv. *oryzae* and 0.90 to 1.45 μg/mL towards *Xanthomonas axonopodis* pv. *citri*. The target QACs bearing a bulky *N*-methylmorpholinium pendant were found to be safe for rice plants as indicated in phytotoxicity test trials. Moreover, the methyl derivative **78** (R = C_6_H_4_-4-Me, *n* = 12), at the dose of 200 μg/mL, applied in vivo in rice bacterial blight disease bioassays afforded control efficiencies of 55.95% and 53.09% in curative and protective activities, respectively. Preliminary mechanism studies performed via SEM imaging suggested that the proposed QACs interact strongly with the bacterial cell membrane [2].

El Achouri and coworkers designed hydroxyalkyl QACs **79** comprising a panel of six novel compounds synthesized through the reaction of aminoalcohols with bromoalkanes containing 12 or 14 carbon atom chains (Figure 49). The derivatives displayed an activity increase as a function of the lengthening of the alkyl substituent against three bacterial strains, *S. aureus*, *E. coli*, and *P aeruginosa*. The antibacterial effectiveness followed this order: CH_2_CH_2_OH > CH_2_CHOHCH_3_ > CH_2_CH_2_CH_2_OH. The lowest MIC values were observed for the G+ pathogen and ranged from 16–130 µM. Compound **79** with a dodecyl and 2-hydroxyethyl fragments proved to be the most active derivative against this strain. The analogical congener **79** bearing a tetradecyl chain was found to be the most effective against Gram-negative species [59].

Epoxy-based QACs were the subject of interest for Li and coworkers in the synthesis of antibacterial compounds [60,136]. Hence, a series of epoxy-functionalized QA surfactants, **80a**–**e**, featuring various alkyl chains (C10–C18), were prepared and evaluated in terms of their antimicrobial action (Figure 6). The decyl derivative **80a** was found to be the most efficient antibacterial agent among the series, exhibiting MIC values against both G+ (*E. coli*, *P. aeruginosa*, *Salmonella*) and G− (*Proteus*, *S. aureus*, *B. subtilis*) strains in the range of 32–80 µM (Table 1) [60].

Yan et al. investigated the impact of counterions on the antibacterial effectiveness of QACs. They synthesized dodecyl- and tetradecyltrimethylammonium derivatives **81** by reacting dimethyl carbonate with corresponding amines. In this reaction, the anion in the salts was replaced with organic acids of varying chain lengths (see Figure 50). Bacteriostatic activity was assessed against *S. aureus* and *E. coli* at a concentration of 90 µg/mL. All compounds inhibited >90% of *E. coli* growth. However, the efficacy against the G+ pathogen gradually decreased with the increase in the carbon chain length of the counterion. QACs with shorter alkyl counterions, especially those not exceeding six carbon atoms, exhibited greater antibacterial efficacy. Formate derivatives **81** (A = HCOO, R = C_12_H_25_ or C_14_H_29_) were identified as the most effective; however, their cytotoxicity was not evaluated (Table 2) [93].

Soukup and colleagues conducted a study on *N*-hydroxyalkyl monoquaternary ammonium salts. The chemical procedure involved a S*_N_*2-type nucleophilic attack, where the amine nitrogen atom attacked the terminal sp^3^ carbon atom of bromoalkanes. The optimal reaction conditions identified a polar aprotic solvent, specifically acetonitrile, as the most suitable for achieving satisfactory yields (Figure 51). The obtained QACs varied in alkyl chain length, with options of 12, 14, or 16 carbon atoms. The antimicrobial efficacy was tested against aerobic G+ bacteria (three strains) and G− bacteria (six strains), anaerobic spore-forming *Clostridium difficile* (one strain), yeasts (four strains), filamentous fungi (four strains), and enveloped Varicella zoster virus. The novel compounds exhibited versatile bioactivity across the entire spectrum of microbes, surpassing the effectiveness of standard benzalkonium salts used as reference antimicrobials in the study. For instance, compound **83c**, substituted with a C_16_ alkyl chain, demonstrated the highest activity as an individual agent against bacteria (Table 1), exceeding the antibacterial activity of commercially used antiseptics. It is noteworthy that **83c**, especially when substituted with the C_16_ alkyl chain, showed exceptional potency against G− strains, such as *Acinetobacter baumannii* (ESKAPE panel), with a minimum inhibitory concentration as low as 1.95 µM. Compound **85a** exhibited a significant reduction in the CFU number of resistant *Clostridium difficile* at 0.5%, while **85c** displayed pronounced activity at a remarkably lower concentration of 0.05%, outperforming benzalkonium salt. Derivative **83c** proved to be the most effective against *P. chrysogenum* filamentous fungi (Table 3), and compound **84b** stood out for its enhanced action against yeasts. Moreover, **84b** showed virucidal potency with a log10 reduction factor of 5. Quantitative Structure–Activity Relationship (QSAR) analysis revealed that activity is higher for large, lipophilic, nonpolar molecules. The investigated compounds were also tested as formulated mixtures to evaluate their potential application as preparations covering the full spectrum of pathogens. At a high concentration of 0.1%, these mixtures demonstrated good practical effectiveness, completely eradicating bacterial and yeast strains. In a skin irritation test using reconstructed human epidermis, the mixtures were found to be safe. Cytotoxicity evaluation conducted in vitro on mammalian CHO-K1 cells confirmed that the elongation of the alkyl chain enhances cytotoxicity, presumably due to growing lipophilicity, correlating with the capability to penetrate cells easily. Notably, the least cytotoxic congeners were pyrrolidinium (**82**) and morpholinium (**85**) [61].

Mancuso and coworkers obtained acryloxyalkyltriethylammonium bromides **86a**–**d** bearing an alkyl chain composed of 6, 9, 11, or 12 carbon atoms via a simple and convenient two-step synthetic procedure from commercially available substrates (Figure 52). The applied method was carried out entirely under air conditions and without the need for chromatographic purification. The antimicrobial activity of all the synthesized QACs was tested against two G+ (*S. pyogenes* and *S. aureus*) strains, three G− bacterial strains (*P. aeruginosa*, *K. pneumoniae*, and *E. coli*), and two yeast strains (*C. albicans* and *S. cerevisiae*). Derivatives **86c**,**d** bearing 11 and 12 carbon atom substituents exerted pronounced antimicrobial action (Table 2 and Table 4), although their activity was relatively weak in comparison to other antibacterials (MICs in the range of 64–1027 µg/mL). The biocidal action was a result of morphological changes and destabilization of the cellular envelope followed by a general increase in outer membrane permeability in a dose-dependent manner [94].

## 3. Antifungal Activity

QACs are attracting increased attention from various research groups due to their noteworthy antifungal properties. Recognizing the potential use of nucleotide analogues in disease treatment, Dmochowska and colleagues embarked on the synthesis of fungistatic QA nucleotides, specifically compound **87**. They achieved this by converting thymidine derivatives into 5′-O-tosylates, followed by displacement of the tosylate group with tertiary amine reagents, namely trimethylamine and trimethylamine (Figure 53). The resultant three compounds underwent in vitro biological testing against *Candida strains*, including *C. albicans*, *Candida glabrata*, and *Candida tropicalis*. Surprisingly, only QAC **87**, lacking an acyl group (R^1^ = H, R^2^ = N(Me)_3_^+^), exhibited a concentration-dependent growth inhibitory effect against two species. The half-maximal inhibitory concentration was observed at approximately 300 μg/mL, with MIC values towards *C. albicans* and *C. tropicalis* measured at 625 and 2500 μg/mL, respectively (Table 4). It is important to note that this activity, while significant, was notably lower than that of the reference drugs fluconazole and amphotericin B [100].

Jain et al. proposed that cationic lipooxazoles could serve as a novel pharmacophore in antifungal drug design. They synthesized 16 molecules, each comprising a quaternary ammonium unit with twin 6–13 carbon chains (Figure 54). These compounds were then assessed for their efficacy in inhibiting the growth of *Candida*. Remarkably, heptyl and octyl derivatives exhibited significant activity, with IC_50_ values ranging from 1.3 to 3.1 μg/mL. Among these, the octyl derivative **88c** demonstrated the highest effectiveness (Table 4). This derivative showed potent activity at relatively low concentrations (MIC 1.6–6.2 μg/mL) against both sensitive and resistant clinical *C. albicans* isolates, as well as non-albicans *Candida* strains, even those resistant to fluconazole. Furthermore, compound **88c** inhibited cell adhesion to surfaces and restricted biofilm formation by 80% at a concentration of 12.5 μg/mL. SEM imaging studies were conducted to elucidate the mode of action. The results confirmed that the active derivative **88c** induced distortion of the *Candida* cell membrane. Fungicidal activity was attributed to membrane destabilization, leading to morphological deformation, including cell shrinkage, collapse, and complete lysis at a concentration of 0.8 μg/mL. Toxicity studies indicated that the hemolytic activity (HC_50_ of 7.4 μg/mL) and cytotoxicity in mammalian cells (IC_50_ of 13.1 and 41.2 μg/mL for human prostate cancer DU-145 and NIH-3T3 mouse fibroblasts, respectively) were higher than therapeutic doses. Despite this, the obtained cationic lipids were deemed suitable for generating broad-spectrum anti-Candida action against both planktonic cells and biofilms. They were also effective against drug-resistant isolates [101].

Wieczorek and colleagues focused on *Candida* species in their research, developing novel long-alkyl gemini QACs **90a**,**b** (Figure 55) and assessing their antifungal properties. The congener **90a**, featuring double 12-hydrocarbon chains, exhibited exceptional efficacy against both drug-susceptible and resistant *C. albicans*, as well as other human or plant fungal pathogens, with minimal inhibitory and fungicidal concentrations at 1 or 2 μg/mL (Table 4). Furthermore, biofilm formation decreased by over 50% following treatment with QACs at 4 μg/mL, and the yeast-to-hyphal transition capability was disrupted. SEM observations revealed morphological changes, such as wrinkled cell surfaces or even cell breakage, at QAC concentrations of 2 μg/mL. To identify potential pathways affected by QAC, RNA sequencing of a mutant library was conducted. The mode of action was reportedly associated with iron homeostasis, deduced from up-regulated genes primarily involved in iron ion transmembrane transport, ferric-chelate reductase activity, oxidizing metal ions, and oxidoreductase activity. Cytotoxicity assessments against human neuroblastoma SK-N-SH and HEK-293 cell lines revealed moderate values, with IC_50_ values of 6.78 and 10.05 μg/mL, respectively. In summary, this QAC with broad-spectrum activity against pathogenic yeasts and filamentous fungi has the potential to disrupt cell growth, hyphal formation, and biofilm development, as well as interfere with iron ion regulation. For these reasons, it may serve as a promising starting point for the development of novel drugs [102].

Rosin, also known as green petroleum, is a raw material derived from softwood. Novel antifungal GS **91** was synthesized from natural gum rosin and its bioactivity was tested against seven fungal strains. The synthetic process involved several steps: first, ethyl rosinate was produced from ethanol and rosin. Subsequently, it was reacted with fumaric acid, and concurrently, an epoxy quaternary ammonium salt was prepared from triethylamine and epoxy chloropropane. In the final step, the bisquaternary ammonium salt of rosinate **91** was synthesized from the two intermediates (Figure 56). The antifungal activity of GS **91** demonstrated effectiveness against various wood decay and mold fungi, with notable efficacy observed against *Chaetomium globosum*, *Phanerochaete chrysosporium*, and *Trametes versicolor*. This new surfactant, derived from renewable resources, is expected to find applications in the industry [137].

Eight QACs (**92a**–**h**) derived from β-pinene were synthesized and screened for their antifungal properties against eleven phytopathogenic fungi by Feng et al. The authors asserted that these novel compounds, originating from essential oils, are anticipated to be more biocompatible, environmentally friendly, and less toxic compared to pure synthetic pesticides. The compounds were synthesized from hydronopyl diethyl amine in reactions with alkyl halides (Figure 57). Overall, the derivatives demonstrated intermediate to high efficacy against the tested plant pathogens at a concentration of 500 µg/mL, with the most promising compounds being **92d**,**e**. These compounds exhibited broad-spectrum activity and inhibitory rates higher than the reference substance, chlorothalonil. The amyl and decyl QAC-pinene conjugates are expected to be valuable for the development of botanical pesticides [138].

Zhang et al. proposed antimicrobial and anticancer quaternary ammonium compounds (QACs) derived from β-pinene. In brief, secondary amines reacted with β-pinene halides to synthesize the desired compounds **93a**–**f** (Figure 58). The obtained molecules were tested against a panel of bacteria, fungi, and cancer cell lines. The investigation revealed that the most successful conjugates exhibited remarkable broad-spectrum antimicrobial potential. Specifically, compound **93a** demonstrated excellent antifungal activity, with IC_50_ values ranging from 4.50 to 33.76 µg/mL, and antibacterial activities, with MIC values ranging from 0.625 to 2.5 µg/mL (Table 2). Noteworthy conjugates **93a**–**c** also displayed activity against cancer cells, with IC_50_ values ranging from 1.10 to 11.63 µM. The most potent compound, **93c**, showed significant anticancer activity, with IC_50_ values of 1.10 µM against human colorectal carcinoma HCT-116 and 2.46 µM against human breast cancer cells MCF-7 (Table 5). Moreover, the examination of relative electric conductivity and morphology revealed that conjugate **93a** caused a reduction in membrane integrity, consistent with the molecular mechanism of action associated with QACs [95].

**Table 5 ijms-25-04649-t005:** IC_50_ values of the discussed QACs against the selected cancer cell lines and non-cancerous HEK-293 as a measure of selectivity [µM].

Compound	A549	MDA-MB-231	HCT-116	HepG2	MCF-7	HEK-293	Ref.
**93c**	-	-	1.1	-	2.46	5.08	[95]
**96a**	-	-	1.04	-	1.53	-	[139]
**97a**	46	-	-	-	-	107	[140]
**100**	2.42	-	1.03	-	-	10.11	[13]
**102d**	-	3.93	-	8.86	4.67	-	[15]
**103a**	-	-	-	130.5	-	-	[62]
**104**	-	4.4	-	-	10.20	-	[14]
**105c**	-	-	-	-	5.5–10.5	-	[141]
**109b**	-	-	-	-	4.9	3.0	[142]

- not tested.

## 4. Anticancer Activity

Cancer is a complex disease characterized by uncontrolled cell division and invasion of abnormal cells. Accordingly, scientists worldwide are actively working to discover a selective cure for this illness. For instance, chondrosarcoma therapy poses a challenge and typically necessitates radical surgery due to the limited effectiveness of radio- and chemotherapy [143]. The delivery of drugs is hindered by the poorly vascularized and dense chondrogenic extracellular matrix. Attaching QAC to the drug may prove beneficial in the selective delivery of cytotoxic agents to the cartilage tumor tissue and facilitate their accumulation, given that QA agents demonstrate a high affinity to negatively charged moieties. Aggrecan, the most abundant proteoglycan in the matrix of chondrosarcoma, is rich in sulfate and carboxylate groups found in chondroitin and keratin sulfate [143]. A research group from Clermont Auvergne University synthesized a QA hybrid agent, **94**, by exhaustively methylating doxycycline with methyl iodide (Figure 59). Compound **94** was tested against matrix metalloproteinases (MMPs), proteolytic enzymes highly expressed in cartilage neoplasm cells, playing a crucial role in cartilage degradation and cell–cell interactions, as well as tumor survival, progression, dissemination, and metastasis [144]. The conjugate **94** demonstrated activity against MMP1 and MMP13 collagenases at IC_50_ of 4.6 and 4.1 µM, respectively, exhibiting higher activity than the parent drug, doxycycline. The compound was evaluated in the Swarm rat chondrosarcoma model in vivo, and tumor growth inhibition was observed at day 12, six days earlier than in the case of unmodified doxycycline. Furthermore, undesirable blood effects associated with the treatment did not occur, and leukocyte decrease was significantly reduced compared to the parent drug. Although the compound was additionally tested for its antibacterial properties, the results did not show any significant activity (Table 2). The disclosed results support the potential application of the innovative agent in diminishing the quantity, dimensions, and pace of metastatic nodule development in animals [96].

The extracellular matrix of chondrosarcoma forms a physical semi-permeable barrier that not only hinders the access of cytotoxic agents to their target but also induces severe chronic hypoxia—a well-known characteristic in tumor progression and a negative prognostic factor associated with chemo- and radio-resistant cancers [145]. Consequently, in the subsequent phase of the study, the research group conceptualized and synthesized a series of novel conjugates comprising QACs and hypoxia-activated prodrugs (HAPs). Typically, HAPs feature a nitro(hetero)cyclic group that serves as a trigger, releasing the cytotoxic drug upon bioreduction under hypoxic conditions. The target hybrids, labeled as **95**, were synthesized through the quaternization of tertiary amine functions using alkyl iodides, as outlined in Figure 60. Their binding affinity to aggrecan was assessed using the surface plasmon resonance technique, while in vitro bioreductive activation was analyzed through an enzymatic reduction assay with *E. coli* nitroreductase.

The experiments confirmed that the designed molecules exhibited rapid bioreductive activation and a high affinity for polyanionic proteoglycan. Testing against the HEMC-SS chondrosarcoma cell line revealed IC_50_ values in the low micromolar range of 0.4–2.8 µM under hypoxic conditions. Notably, the conjugates displayed selectivity versus normoxia. Specifically, compound **95** with a butyl linker (*n* = 4) and benzyl, ethyl, and methyl substituents in the R^1^, R^2^, and R^3^ positions, respectively, demonstrated the highest hypoxia cytotoxicity ratio (HCR) of 24 [146]. Continuing the investigation, the researchers introduced alterations in the positions of the proteoglycan-targeting ligand and modified the structure of the alkylating mustard. Additionally, attempts were made to replace the aromatic ring in the hypoxia-sensitive trigger from diazole to furan or benzene; however, none of these modifications resulted in improvements in selectivity toward hypoxic cells [147].

Omran et al. reported the synthesis of phenanthroindolizidine hypoxia-targeted prodrugs **96a**–**c** through the reaction of tylophorine or antofine with bromomethylnitroimidazole (Figure 61). The polar conjugates **96a**–**c** exhibited a notable, up to 80-fold increase in water solubility compared to the lipophilic precursor alkaloids. Additionally, their predicted ability to cross the blood–brain barrier (BBB) was diminished, suggesting a potential reduction in neurotoxic side effects associated with more hydrophobic parent compounds. The hybrids were tested against five malignant and two noncancerous cell lines. While the novel derivatives exhibited remarkably low IC_50_ values, their cytotoxic activity was comparatively lower than that of the parent alkaloids (Table 5). However, there was a notable enhancement in selectivity for hypoxic tumors, reaching a HCR of up to 26 for **93a** [139].

Lubeau and coworkers designed and synthesized cationic alkylphospholipid prodrugs for antitumor therapy. Initially, they obtained QA alkylphospholipids, including miltefosine, perifosine, and erufosine. Subsequently, direct O-alkylation was employed to produce alkyl and alkyloxy ester derivatives **97**–**99** (Figure 62). The hydrophobic modification significantly enhanced the biocompatibility of the prodrugs, leading to a substantial reduction in their haemolytic activity compared to the parent compounds. The antiproliferative activity of these prodrugs was evaluated against two cancer and one non-cancerous cell lines, revealing IC_50_ values in the micromolar range, closely resembling those of the parent compounds (Table 5). Additionally, the authors suggested the potential application of these compounds in gene therapy, as they demonstrated the ability to form complexes with plasmid DNA. Particularly, derivatives of erufosine exhibited effectiveness in transfecting DNA, encoding the pro-apoptotic protein tumor necrosis factor-related apoptosis-inducing ligand (TRAIL) into mammalian cells. Consequently, these prodrugs could serve as carriers in gene delivery. This innovative approach has facilitated the development of selective and well-tolerated agents suitable for intravenous injection, offering promise in the treatment of drug-resistant tumors [140,148].

Quaternization is frequently employed to enhance the water solubility of compounds. This strategy was utilized by Yang and colleagues to modify the pharmacokinetic properties of diosgenin, a natural steroidal saponin with diverse bioactivities, including antitumor action. They synthesized 22 novel diosgenyl derivatives, designated as **100** and **101**, incorporating QAC through the use of amines and bromoester derivatives of diosgenin precursors (Figure 63). The water solubility of these derivatives was improved, and their antiproliferation activity against various malignant cell lines exceeded that of the parent compound. For instance, compound **100** (with R^1^ = R^2^ = *n*Bu) exhibited significantly low IC_50_ values (Table 5) [13].

In another application of quaternization to enhance water solubility and improve bioavailability, Skrzypczak et al. [15] conducted a study involving the synthesis of five novel quaternary *N*-alkyl ammonium derivatives (**102a**–**e**) of the ansa-macrolide geldanamycin. The process involved replacing the methoxyl group of geldanamycin with the amine quinuclidine group, followed by alkylation (Figure 64). The resulting compounds were then evaluated against nine cancer and two normal cell lines. Most of the compounds exhibited activity in the low micromolar range (IC_50_ 1–10 µM), with compound **102d** demonstrating broad-spectrum activity against all tested cancer cell lines (Table 5). Notably, these molecules demonstrated heightened potency against the MCF-7 cell line, with reduced toxic effects in normal human lung fibroblasts CCD39Lu (IC_50_ > 10 µM). This effect is attributed to the increased water solubility and reduced lipophilicity of the synthesized molecules. Molecular docking studies further revealed that the proposed agents interacted with their molecular target, heat shock proteins Hsp90, through additional hydrophobic interactions facilitated by the amine moiety at the C17 position [15].

Karataş and coworkers synthesized three coumarin-based tetra-alkyl ammonium salts (**103a**–**c**) through a coupling reaction of a 4-chloromethylene coumarin substrate with tertiary amines, as illustrated in Figure 65. The goal of this modifications was to enhance water-solubility and biological activity of the parent structure. Hence, the novel compounds (**103a**–**c**) were then assessed for their cytotoxic properties against human liver cancer HepG2 and Caco-2 cell lines. The results indicated that the synthesized compounds exhibited low cell toxicity, comparable to the standard drug cisplatin. However, when tested against the non-cancer mouse fibroblast cell line L-929, their activity was significantly lower than the reference (from 224 to above 350 vs. 74 µM), as shown in Table 5. Furthermore, all compounds (**103a**–**c**) demonstrated potent inhibition of human carbonic anhydrase I and II, as well as xanthine oxidase, outperforming the standard drugs acetazolamide and allopurinol, respectively. Although the compounds were evaluated against *E. coli* and *B. subtilis*, they were found to be inactive in antibacterial studies (Table 1) [62].

Longo and coworkers proposed the introduction of QAC to enhance the activity of ellipticine, i.e., an alkaloid withdrawn from metastatic breast cancer treatment due to poor solubility in water and severe side effects. The researchers designed and synthesized a series of quaternary ammonium salts **104** by incorporating *N*-alkyl substituents into the carbazole moiety, followed by exhaustive alkylation with methyl iodide (Figure 66). The newly synthesized compounds were evaluated for their ability to inhibit human topoisomerase II using various methods, including in silico and in vitro approaches. This enzyme, overexpressed in tumor cells, plays a crucial role in DNA transcription and replication. The results showed a significant improvement in the solubility of the compounds, coupled with noticeable cytotoxic activity against breast tumor cells. This activity was attributed to enzyme inhibition and the induction of apoptotic cell death. Importantly, none of the tested derivatives affected normal breast cells (MCF-10A), with an IC_50_ greater than 500 µM. The most potent chlorine derivatives (**104**) with 5- or 6-carbon chains demonstrated effectiveness against two breast cancer cell lines, MCF-7 and the highly metastatic and aggressive MDA-MB-231. The active compounds exhibited low micromolar IC_50_ ranges of 10.2–20.4 µM and 4.4–5.8 µM for MCF-7 and MDA-MB-231, respectively. The compound with the heptyl linker (R = Cl, *n* = 5) showed the highest potency among them (Table 5). These results are particularly promising for addressing the challenges posed by the highly metastatic and aggressive MDA-MB-231 cell line [14].

Previous studies have indicated that mitochondria in carcinoma cells generally feature an elevated transmembrane potential compared to normal cells [149]. This observation has led to the hypothesis that compounds chemically modified with positively charged hydrophobic groups may effectively target these organelles, accumulating in the mitochondrial matrix. Spinosyn demonstrates antitumor action by inhibiting oxidative phosphorylation in mitochondria [150]. Ma et al. detailed a multistep synthesis of 13 quaternary ammonium (QA) spinosyn derivatives, **105a**–**c**, with enhanced anticancer properties achieved through the alkylation of tertiary amine intermediates (Figure 67). Some of the obtained derivatives exhibited significantly improved antiproliferative abilities against four tested human cancer cell lines, with the most active compound being **105c** (Table 5). The assessed lipophilicity was identified as a crucial factor in influencing the antiproliferative effects, demonstrating a correlation with the observed activity. Additionally, compound **105c** (m = 11, *n* = 0) displayed notably greater oxidative phosphorylation inhibition and apoptosis-inducing ability compared to the parent compound, spinosyn A [141].

Roayapalley et al. implemented a similar approach, targeting mitochondrial membrane potential using QA-conjugated dienones **106a** as multifunctional ligands and candidates for antineoplastic agents (Figure 68). Evaluation against oral squamous cell carcinoma (Ca9-22, HSC-2, and HSC-4 cells) revealed that the QA compounds exhibited potent cytotoxicity with submicro- and nanomolar IC_50_ values in virtually all cases, comparable to doxorubicin and superior to the precursor oximes and the reference compound melphalan. In contrast, the conjugates demonstrated lower cytotoxicity towards non-malignant human oral cells (gingival fibroblasts HGF, periodontal ligament). Specifically, 4-chloro **106c** and fluoro **106b** lowered the mitochondrial membrane potential in CEM cells, and in turn, 3,4-dichloro **106d** induced transient G2/M accumulation in Ca9-22 cells. Three compounds, i.e., **106c**–**e**, were identified as lead molecules with drug-like characteristics [151].

The *Safirinium* compounds are highly fluorescent molecules with adjustable lipophilicity and emission wavelength, developed by Sączewski et al. as easily prepared, low-cost dyes [152,153,154,155,156]. These QACs can be readily conjugated with nucleophilic chemical species, making them suitable for fluorescent labeling in biochemical studies [155,156]. Csuk and coworkers combined fluorescent *Safirinium* compounds with acetylated triterpenoic acids (Figure 69), aiming to create dual-functionality compounds **107**–**109** as cytotoxic molecular probes. The obtained hybrids exhibited moderate toxicity, except for betulinic acid compound **109b** (Table 5), which showed an IC_50_ ranging from 4.6 to 7.5 µM for ovarian carcinoma A2780 and colorectal adenocarcinoma HT29 cells, respectively. However, the terpenoic QACs lacked selectivity overall. Surprisingly, a fluorescence imaging investigation of compound **109b** revealed that the conjugates accumulate in the endoplasmic reticulum but cannot enter the mitochondria within the limits of detection [142].

Photodynamic therapy (PDT) requires the use of a proper wavelength of light to stimulate the photosensitizer on the lesion area, offering selective destruction of the target while avoiding unwanted damage to healthy tissues [157]. Tang and collaborators proposed multicationic molecules end-capped with QAC as a new generation of photosensitizers for photodynamic theranostics. They synthesized compound **110**, tethered with four quaternary ammonium heads (Figure 70).

Compound **110** was tested in PDT against the HeLa cancer cell line and demonstrated the ability to inhibit 50% of the cells at concentrations of 2.5–5 µM, attributed to its capacity for singlet oxygen generation under white light illumination. In contrast, the compound showed no activity and did not affect cell viability in the absence of electromagnetic radiation. Furthermore, the ammonium terminal-functionalized photosensitizer **110** exhibited fair binding affinity with DNA. Fluorescence spectroscopy experiments revealed its predominant localization in the mitochondria of living cells, while displaying a non-rapid cytoplasm-to-nuclei transition in dead cells. The reported results underscore the significant potential of the novel agent **110** in photodynamic theranostics [158].

Another investigation involved the synthesis of a range of quinizarin derivatives incorporating QACs, which were subsequently evaluated for their anticancer properties against leukemia cell lines. Quinizarin reacted with 1,4-bis(bromomethyl)benzene, followed by reaction with tertiary amines to yield mono-derivative **111a** and bis-QAC derivatives **111b**,**c** (Figure 71). Compound **111a**, when reacted with triethanolamine, produced compound **111c**. The synthesized molecules demonstrated inhibition of the proliferation of various leukemia cells, with the most potent compound being **111a**, showing IC_50_ values between 1.40 and 3.51 μM. Moreover, compound **111a** induced significant apoptosis in Molt-4 and Jurkat cells in a dose-dependent manner. Cell cycle analysis in the presence of conjugate **111a** revealed proliferation inhibition in the mentioned cell lines, accompanied by a marked G0/G1 phase arrest. Additionally, a considerable decrease in the S-phase cell population was observed in these cell lines. Compound **111a** exhibited the ability to generate reactive oxygen species (ROS) in the treated cells. Western blotting assays demonstrated its activation of the caspase-dependent apoptosis pathway and induction of the degradation of Bcl-2 and c-myc proteins. These factors are associated with chemotherapy resistance in acute T lymphoblastic leukemia [159].

Poly(ADP-ribose) polymerase 1 (PARP-1) plays a central role in controlling androgen receptor function in prostate cancer cells. NAD-like PARP-1 inhibitors used in clinical practice cause unwanted off-target effects due to a lack of high selectivity. Karpova et al. prepared ester and amide analogues of cyclododecyl and methylated cyclic aliphatic amines **112a**–**f** (Figure 72) as novel PARP-1 inhibitors effective for the treatment of this disease state. The synthesized QACs were not structurally related to commercial drugs; consequently, they could be superior to classical inhibitors. Experiments against androgen-dependent (LNCaP) and castration-resistant (PC-3 and DU-145) cell lines revealed that the most potent compounds were ester aminoalkyl derivatives. Compound **112a** was found to be active against both androgen receptor-positive and -negative cancer cells susceptible to PARP-1 inhibition. The authors suggest that these inhibitors may be effective therapies for patients with castration-resistant prostate cancer [160].

## 5. Various Applications

### 5.1. Antivirals

The incorporation of a quaternary ammonium moiety may prove beneficial in the development of antiviral agents. Sokolova et al. focused their research on the influenza A virus. They designed and synthesized novel QACs **113a**,**b** and **114a**–**d** through the methylation of *N*,*N*-dialkyl (+)-camphor imines or (−)-borneol derivatives (Figure 73). Notably, the most active borneol-QAC conjugate, **114a** (*n* = 1, R = Me), demonstrated superior selectivity. Conjugate **114a** exhibited activity against influenza A viruses with an IC_50_ ranging from 2.4 to 16.8 µM, while its IC_50_ value in MDCK cells was 1311 µM, resulting in a selectivity index (SI) ranging between 78 and 546. In contrast, the IC_50_ values of ribavirin, the commercial antiviral agent used as a control in the study, ranged between 24.6 and 41 µM, indicating its inferior efficacy compared to the novel compound. Subsequent hemolysis assays revealed that conjugate **114a** significantly reduced the membrane-disrupting activity of the influenza A virus. Consequently, it could be considered a promising candidate as an anti-influenza membrane fusion inhibitor [161].

Phenanthroindolizidine alkaloids were not only utilized as anticancer chemotherapeutics but also as demonstrated efficacy as antivirals. Wang and collaborators directed their efforts towards the tobacco mosaic virus (TMV), an infectious agent causing significant economic losses in crops such as tobacco, tomatoes, peppers, cucumbers, and various other plants. The quaternization of tylophorine with various alkyl bromides, carrying activating groups in the β position, was executed (Figure 74) to address the alkaloid’s low stability, enhance its water solubility, and reduce its ability to penetrate the blood–brain barrier (BBB), thus mitigating central nervous system (CNS) toxicity. In general, all the QAC-(*R*) **115** and (*S*) **116** tylophorine conjugates exhibited promising in vitro and in vivo antiviral activity, with some surpassing the efficacy of the parent compound and ribavirin. Among the tested compounds, trans (*S*)-propargyl quaternary ammonium salt **116g** demonstrated the most potent anti-TMV activities. Notably, the solubility of this compound was over 500-fold higher than that of its parent compound, and its stability was significantly improved. These findings offer valuable insights into the development of plant virus inhibitors, and the obtained structure **116g** is expected to be a viable alternative to currently employed agents [162].

### 5.2. Antiprotozoals

The activity of QACs was also evaluated against parasitic diseases. López-Muñoz et al. synthesized compounds with a terminal system of two phenyl rings bearing fluorine substituents (**117**) and tested them as potential trypanocidal agents. They hypothesized that the presence of a fluorine atom would impact the bioactivity of the molecules. In the multi-step synthesis, they obtained amine derivatives and converted them into the target QACs **117** using either iodomethane, diiodomethane, or chloroiodomethane (Figure 75). Compound **117** (X = I, *n* = 4, R = C_6_H_4_-3-CF_3_) exhibited the highest antitrypanosomal activity (IC_50_ value of 0.9 μM); however, this derivative was the most toxic in the cytotoxicity assessment on promonocytes U-937. In contrast, QAC **117** (X = I, *n* = 4, R = C_6_H_4_-4-F) was the most selective, exhibiting an SI of 30.4, making this derivative the most promising compound. Some general trends were observed; for example, QACs bearing an *N*-iodomethyl substituent were more effective than those with an *N*-chloromethyl group. Additionally, compounds with a longer tether (*n* = 3 > *n* = 2) presented higher activity. On average, monofluoro-substituted aromatic rings gave rise to a more desirable activity profile in terms of high SI value and low toxicity. On the contrary, trimethylfluoro substitution resulted in greater cytotoxicity. Furthermore, para-substitution tended to render improved antiprotozoal QACs [163].

### 5.3. Analgesics, Anesthetics, and Muscle Relaxants

Pain sensation can be reduced by inhibiting the activity of nociceptor neurons responsible for initiating and transmitting action potentials through the application of local anesthetics. However, these therapeutics may inadvertently affect motor neurons and other sensory functions, leading to significant adverse effects, especially with prolonged use. To mitigate the toxicity and safety concerns associated with traditional pharmacotherapies, the utilization of photosensitive drugs capable of rapid and reversible activation with light has been explored. The ability to modulate the biological activity of isomerizable compounds through electromagnetic irradiation at various wavelengths allows precise control over their temporal and spatial effects. A promising development in this area is the optopharmacological agent incorporating a chemical photoswitch, designed and synthesized by Mourot and colleagues. Their objective was to create a light-sensitive local anesthetic with a specific impact on nociceptors. The resulting compound, labeled as **118**, shared structural and functional similarities with lidocaine but possessed the added advantage of controllability through irradiation. The novel gemini QAC **118** was synthesized through a reaction sequence involving an azobenzene derivative, as outlined in Figure 76. Subsequent characterization of its photo-pharmacological properties revealed its potency as a blocker of native intrinsic K^+^ and Na^+^ channels in primary sensory neurons in the trans form. Conversely, in the *cis* configuration, it relieved the blockade and restored the excitability of pain-sensing neurons, as evidenced by the recurrence of channel activity.

The currents were more pronounced under blue light compared to darkness, where neuronal action was inhibited. Only a single wavelength of light was required to rapidly toggle in and out of its binding site on ion channels, since the reverted relaxation process back to the *trans* state occurred spontaneously within seconds after light was shut off. The compound diffused readily through native tissue within minutes, reaching all cells susceptible to photosensitization and accumulating in neurons through activated nociceptive ion channels. It could selectively target hyperactive neurons, silencing those in need the most and providing pain-selective analgesia. The blockade of voltage-gated ion channels, present in all neurons, could be finely tuned with light wavelength and/or intensity. This research offers a valuable non-invasive method to photo-control pain signaling with fast kinetics and high selectivity. The pioneering compound is a highly soluble small molecule, making it an ideal candidate for long-lasting optical control of action potential firing and treating chronic pain while simultaneously minimizing unwanted side effects in other neurons [164].

Rodríguez-Franco’s research group designed a series of novel photoswitchable neuromuscular ligands by incorporating two *N*-methyl-*N*-carbocyclic quaternary ammonium moieties into an azobenzene core at the m- or p-position (Figure 77). The resulting ligands, similar to **118**, were named azocuroniums **119**. They could be activated and deactivated ad libitum through light-induced photoisomerization between the (*E*)- and (*Z*)-forms. As anticipated, the compounds were highly soluble in physiological media, displayed negligible in vitro toxicity, and were predicted to be CNS-impermeable via passive diffusion. In radioligand binding assays, *meta*-azocuroniums exhibited potent nicotinic blocking activity (except for the pyrrolidine derivative *m*-**119a**, which acted as an agonist), demonstrating clear subtype selectivity towards muscular nicotinic acetylcholine receptors (nAChRs). Derivatives with the smallest cationic heads, namely the pyrrolidine *m*-**119a** and the piperidine *m*-**119b**, emerged as the most potent ligands for muscle-type nAChRs, with binding affinities in the nanomolar range (*K*_i_ = 42 nM and 35 nM, respectively). Simultaneously, they showed much lower affinities for neuronal receptors α7 (*K*_i_ of 2500 and 910 nM, respectively) and α4β2 (*K*_i_s > 10,000 nM), where they were nearly inactive. The synthetic accessibility, coupled with the ability to control their qualitative effects through modification of the quaternary ammonium residues, renders this family of compounds intriguing candidates for the development of light-targeted muscle relaxants for surgical procedures with fewer side effects [165].

Bufotenine is a tryptamine alkaloid initially isolated from secretions of the skin and posterior glands of Asiatic toads Bufo bufo gargarizans Cantor and Bufo melanostictus Schneider. This compound is formed through serotonin methylation and has shown potential in alleviating the severity of pain and related syndromes. The quaternary ammonium bufotenine derivative **120** was synthesized from 5-benzyloxyindole as the raw material (Figure 78). The objective was to evaluate its analgesic activity in vivo. The compound exhibited robust effects on formalin-induced pain behavior in mice, as indicated by a reduction in paw licking/biting. Its activity surpassed that of bare bufotenine. Furthermore, when used in combination with morphine, the quaternary ammonium bufotenine derivative **120** demonstrated a synergistic action in the mouse hot plate analgesia assay. The compound was suitable for short-acting analgesia, as its effect diminished after 1.5 h post-administration. To identify the action mechanism, 36 analgesic-related targets were systemically evaluated utilizing reverse docking. The computational predictions covered structures such as 15 G protein-coupled receptors, 6 enzymes, 13 ion channels, and 2 other potential molecular targets. The study revealed that acetylcholinesterase (AChE) or the α4β2 type of nicotinic acetylcholine receptor (nAChR) met the criteria as the most probable binding sites for this compound [166].

Quadri et al. focused their research on nicotinic acetylcholine receptors (nAChRs), specifically targeting the α7 subtype. This subtype has emerged as a potential target for treating neurological, neurodegenerative, neuropsychiatric, neuroinflammatory disorders, and as a therapeutic approach for pain. The researchers synthesized a novel group of QACs, **121a**–**h**, based on the spirocyclic quinuclidinyl-Δ^2^-isoxazoline (Figure 79). Their goal was to identify potent silent agonists, i.e., derivatives effective at low concentrations and demonstrating activity in the presence of a positive allosteric modulator (PAM). The hypothesis behind their work was that the positive charge in the compounds could lead to strong ionic interactions, inducing diverse desensitization states through conformational changes in the receptor. Various substituents were introduced in the position 3 on the phenyl ring of the benzyloxy moiety. It was speculated that polar interactions could increase protein–ligand stability, and halogen atoms might contribute to halogen bond formation within the receptor. The QACs were assessed for silent agonist activity, and electrophysiological experiments revealed an enhanced action profile for some of them. Among the tested compounds, 3-halo QAC **121b** exhibited the most favorable profile in the entire series. The researchers found that halogen-substituted compounds with specific interactions could stabilize a desensitized conformational state of the receptor, sensitive to a type II PAM. Notably, chlorine was identified as an optimal *meta*-substituent for preferentially inducing a desensitized state, while bulkier groups (such as naphthalene, Br, I) were found to be detrimental to silent agonism. The presence of a permanent positive charge was deemed crucial for promoting silent activation [167].

Del Bello et al. synthesized novel antagonists of the muscarinic acetylcholine receptor (mAChR) and studied their affinity at M_1_-M_5_ receptor subtypes. A multi-step synthetic pathway produced dioxanes, which were treated with methyl iodide and transformed into methiodides **122**–**125** (Figure 80). Subsequently, radioligand binding assays were conducted using five human recombinant members of the mAChR family (hM_1_–hM_5_) expressed in CHO cells. Among the novel QACs, compound **122** with cyclohexyl and phenyl moieties (R^1^ = C_6_H_11_, R^2^ = Ph) in the 6-position showed the most favorable interaction as evidenced by the exceptional inhibition binding constant values and a selectivity profile analogous to that of the therapeutically approved drug oxybutynin. Docking simulations on the resolved structure of human M_3_ mAChR were performed to elucidate the binding mode and rationalize the experimental observations. The M_3_/M_2_ selectivity ratio of 14.5 should prevent cardiovascular adverse effects while the presence of quaternary ammonium function may limit the crossing of the BBB and diminish central anticholinergic activity, hence minimizing CNS side effects. Therefore, the new compound represents a valuable starting point in the development of bioactive molecules potentially useful in peripheral diseases, such as overactive bladder, where M_3_ mAChR receptors are involved [168].

Kracke and coworkers synthesized neuromuscular blocking agents (NMBAs) based on decamethonium by inserting a three-dimensional isomeric carborane cluster in the methylene chain connecting the two quaternary ammonium cations. The structure of super aromatic QACs, icosahedral dicarba-*closo*-dodecaborane molecules **126a**–**c**, is depicted in Figure 7. In animal experiments, the three quaternary ammonium *ortho*-, *meta*-, and *para*-carboranes **126a**–**c** reversibly induced muscle relaxation in grip strength and inverted screen tests in mice. The muscle weakness was followed by a partial (**126a**) or complete (**126b**,**c**) recovery within 30 min. The potency rank was *para* > *meta* > *ortho*. Pharmacologic and electrophysiologic experiments in vitro verified that all the compounds were characterized by a nondepolarizing mechanism of action. The conjugates do not fall within the benzylisoquinoline or steroid classes of NMBAs. Consequently, they offer promising alternatives to drugs presently employed in surgical procedures and various applications, such as cosmetics [169].

### 5.4. Anticholinesterases

Pashirova and coworkers aimed to synthesize hybrid water-soluble anti-cholinesterase pharmacophores containing a sterically hindered phenol moiety. The proposed multi-target QACs could serve as promising templates for the development of anti-Alzheimer drugs. They successfully obtained two novel quaternary ammonium molecules, **127a** and **127b**, through the reactions of previously reported phenol derivatives with benzyl bromide (Figure 81). The structural modifications were anticipated to enhance bioavailability and solubility while reducing toxicity and degradation rates. Compound **127a**, featuring a 2-carbon chain linker, demonstrated higher activity against human butyrylcholinesterase (BChE) compared to its 3-carbon chain analogue, with an IC_50_ value of 20 µM versus 40 µM. Additionally, this conjugate exhibited a 10-fold selectivity towards human AChE. The obtained conjugates demonstrated antioxidant properties and showed no toxicity in hemolytic assays, even at high concentrations. Moreover, they form cationic liposome nanosystems suitable for drug delivery in clinical practice [170].

Topuzyan et al. identified compounds with anticholinesterase activity. They synthesized tertiary aminoamides and subsequently converted them into iodomethylates (Figure 82). In the evaluations conducted on human erythrocytic AChE and plasmic BChE, these conjugates demonstrated exceptional selectivity for the latter enzyme (selectivity index, SI, in the range of 1000–1423). Furthermore, the obtained compound **128** demonstrated outstanding inhibitory properties, with IC_50_ values ranging from 0.11 to 0.78 µM for BChE compared to 180–1430 µM for AChE. In addition, the novel conjugates exhibited moderate activity against both *S. aureus* and *E. coli* [171].

The development of cholinesterase inhibitors was also pursued by Csuk and coworkers. They utilized triterpenoic acids, such as oleanolic, ursolic, betulinic, platanic, and glycyrrhetinic acid, to synthetize the corresponding amide derivatives through direct acylation with 1,3- or 1,4-diazabicyclo[3.2.2]nonanes. Subsequently, these amides underwent reaction with methyl iodide, leading to the formation of salts with a quaternary ammonium moiety at the distal nitrogen of the bicyclic system. The QACs **129**–**133** (Figure 8) demonstrated strong inhibition of BChE while exhibiting poor inhibitory effects on AChE. Particularly, olean-, lupane, and ursane type conjugates (**129**–**131a**) exhibited high specificity for BChE, with inhibition rates exceeding 95% at 10 µM, while AChE activity was only minimally affected (<25%). Molecular docking studies suggested that the variations in cholinesterase selectivity could potentially be explained by polar interactions and steric hindrance in the active site of the enzymes [172].

Padrtova et al. synthesized quaternary ammonium cholinesterase inhibitors through a multistep process starting from *p*-aminobenzoic acid (Figure 83). The synthesis of QACs involved a Menshutkin reaction between tertiary amines and alkyl halides, facilitated by microwave radiation. Initially, nine novel QACs (**134**) were synthesized, featuring the general structure of an arylcarbonyloxyaminopropanol with a carbamate group in the position 4. The physicochemical properties of these compounds, including lipophilicity (calculated using software) and experimentally determined surface tension, were evaluated to estimate their pharmacokinetic parameters. Subsequently, the cholinesterase inhibitory activity of the synthesized compounds was measured. Generally, the compounds exhibited a pronounced tendency to inhibit AChE at 100 µM concentration. Those with bulkier substituents in the basic part of the molecule demonstrated a stronger affinity to both enzymes, although their ability to inhibit BChE was comparatively lower. Notably, among the tested QACs, compound **136** (R^1^, R^2^, and R^3^ = Et, Et, and Me or Pr, Et, and Me, respectively) exhibited the most promising inhibitory activities. In the final phase, a molecular modeling study was conducted to gain insights into the molecular interactions and mode of action of the most active compounds. The reported data provide valuable information regarding the possibilities of structural modification and the enhancement of the biological effects in the design of novel AChE inhibitors [173].

Godinez, Lee, and Schwans explored the impact of acetylcholine depletion on cognitive function in neurodegenerative diseases such as Alzheimer’s. Elevated activity of butyrylcholinesterase (BChE) was implicated in reducing acetylcholine levels in individuals with Alzheimer’s. In an effort to find effective BChE inhibitors to counteract acetylcholine degradation, the study investigated 9-fluorenylmethoxycarbonyl (Fmoc) amino acid-based compounds **135** (Figure 9). The incorporation of a cationic trimethylammonium group to mimic acetylcholine’s structure resulted in the synthesis of a series of inhibitors. While Fmoc-ester derivatives acted as substrates, Fmoc-amide derivatives selectively inhibited BChE (IC_50_ 0.06–10.0 µM), demonstrating enhanced potency. Computational docking studies indicated interactions with the cholinyl binding site and peripheral site [174]

### 5.5. Herbicidals

Wu et al. designed and synthesized novel candidates (**136** and **137**) for weed control, utilizing alachlor and acetochlor as the parent scaffolds (Figure 84). The researchers confirmed that QACs **137**, when combined with acid anion, significantly enhanced the herbicidal action of the molecules. The prepared compounds demonstrated notable herbicidal activities, specifically targeting the control of undesirable plants such as velvet leaf, common zinnia (youth-and-age), barnyard grass, and foxtail. Notably, compound **137b** (R^1^ = Et, R^2^ = H) exhibited excellent results, with an IC_50_ range of 15.9–36.2 g a.i./ha. Field trials indicated that this compound surpassed the commercial herbicide imazethapyr in controlling broadleaf weeds at a dosage of 150 g a.i/ha, exhibiting a killing rate twice as high. However, its efficacy was found to be slightly inferior to other tested species. The proposed QACs can serve as leads for further modifications in the development of crop protection products [175].

Żelechowski et al. synthesized a series of novel QACs containing a terpenoid fragment. Perillyl alcohol, derived from limonene, citronellol, and geraniol served as the starting materials in the synthetic processes. The respective alcohols were transformed into corresponding bromides, followed by the alkylation of tertiary amines, resulting in the production of eighteen new QACs **138**–**140** (Figure 85). The herbicidal and fungicidal activities of the synthesized compounds were evaluated against ten species of temperate climate weeds from the monocotyledon and dicotyledon classes, six plant pathogens, and four fungi responsible for the decomposition of paper and wood. The majority of the conjugates exhibited a broad spectrum of biological activity, with several surpassing the efficacy of the reference compounds. Certain activity dependencies based on logP were discussed. The study suggests that these novel ammonium salts, featuring a terpene-derived moiety, could be considered effective and readily biodegradable preservatives against organisms that contribute to the degradation of wood and technical materials [176].

## 6. Conclusions

In summary, recent reports demonstrate that QACs constitute a promising class of compounds with a wide range of biological activities. The scientific literature presents a plethora of novel quaternary ammonium small molecules obtained through synthetic methods, showcasing their versatile applications as antibacterial, antifungal, and anticancer agents. Beyond these well-established roles, QACs exhibit bioactive properties such as antiprotozoal, herbicidal, analgesic, anesthetic, muscle relaxant, and cholinesterase inhibitory activities. Undoubtedly, future publications will introduce new modifications of QACs, and researchers worldwide will continue their efforts to enhance the biological potencies of these compounds. It should be emphasized that the biological activity of QACs is strongly influenced by the balance between hydrophilicity and lipophilicity [177]. In general, it has been proven that quaternary ammonium compounds (QACs) with varying chain lengths exhibit distinct effectiveness in different applications due to their diverse physicochemical properties. For example, research has demonstrated that a long alkyl chain substituent, typically comprising at least eight carbons, enhances the antimicrobial activity of QACs. However, as indicated in this review, the optimal alkyl chain length varies depending on the specific structure of the QAC. Despite numerous years of research in this field, scientists in medicinal chemistry continue to pursue optimal derivatives through trial and error [178]. The utilization of experimental methods [179] or computational approaches [180] to determine the ideal lipophilicity of the designed active substance remains a distant prospect.

## Data Availability

No new data were created or analyzed in this study. Data sharing is not applicable to this article.

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
