# Peer review of "Advances in the Synthesis of Biologically Active Quaternary Ammonium Compounds"

_ijms, 2024, doi:10.3390/ijms25094649_

Round 1

Reviewer 1 Report

Comments and Suggestions for Authors

I carefully reviewed this manuscript. I didn’t find important scientific or technical errors and mistakes. There are following grammatic comments or suggestions. If the author(s) agree with it, correct each point. If you make some of the comments, this manuscript will be published in this journal. I smoothly reviewed this manuscript.

L12: “the review” is changed to “this review”.

L23-29: This sentence is too long. It is better to separate into two or more sentences.

L24: For “NR4+”, subscript and superscript are used.

L34: It is better to change “In the present work” into “In this review”.

L80: For the abbreviation of “logP”, more explanation is required.

L113: “cells, therefore” is changed to “cells. Therefore”.

L133: A compound “1c” is not shown in any scheme.

L154: What does mean the numeral “1”?

L173: It is better to change “form occurs” into “form is formed”.

L181: “AMP” is changed to “AMPs”.

L282: The phase of “18b able to inhibit” is modified.

L314: “in Table 1. [41].” Is changed to “in Table 1 [41].”

L331: “et al” is changed to “et al.”.

L423: “, however, it did not” is changed to “. However, it did not”.

L439: In this manuscript, the terms of review, work, and study are mixed. It is better to unify the use of term, for example, “review”.

L492: “et. al” is changed to “et al.”.

L498-500: This sentence should be modified for readers to enhance the understanding.

L539: “42a, however, due to” is changed to “42a. However, due to”.

L577: “GS were” is changed to “GSs were” or “GS was”.

L605: Is “GSs” correct?

L665: “Table 2 and 4” is changed to “Tables 2 and 4”.

L704: The size of Scheme 35 is a little expanded.

L745: “This research team” Does it refer to which team?

L761: “Table 1 and 3” is changed to “Tables 1 and 3”.

L767: “These environmentally friendly molecules” is better.

L793: “Table 1 and 3” is changed to “Tables 1 and 3”.

L805: “[94, 95]” is changed to “[94,95]. Remove a space between 94, and 95.

L883: “hydroalocoholic solution” is changed to “hydroalocoholic solutions”.

L893: What was irradiated?

L940: The size of Scheme 45 is a little expanded.

L1004: There are two spaces between fragments and proved. One space is removed.

L1020: “In the reaction” is changed to “In this reaction”.

L1073: “Table 2, 4” is changed “Tables 2 and 4”.

L1101: The term “Table 4” is crosslined.

L1121: There are two spaces between The and congener. One space is removed.

L1126: The abbreviation of SEM is already used.

L1168: It is better to change “were reacted with” to “reacted with”.

L1201: “MMP” is changed to “MMPs”.

L1205: The abbreviation “IC50” is already used.

L1221: “HAP” is changed to “HAPs”.

L1404: It is better to change “were reacted with” to “reacted with”.

L1457: “but also demonstrated” is changed to “but also as demonstrated”.

L1555: “This molecule” is changed to “This compound”.

L1621: Here is “min”. The term “minute(s) is also used. In the term “hour(s) is also used. The unit of time should be unified.

L2041: Only this journal name is abbreviated.

Comments on the Quality of English Language

I smoothly read and review this manuscript. After some revisions, this manuscript is suitable to be published in this journal.

Author Response

We thank the reviewer for the detailed inspections, which have led to significant improvements of the manuscript.

Reviewer 1

I carefully reviewed this manuscript. I didn’t find important scientific or technical errors and mistakes. There are following grammatic comments or suggestions. If the author(s) agree with it, correct each point. If you make some of the comments, this manuscript will be published in this journal. I smoothly reviewed this manuscript.

L12: “the review” is changed to “this review”.

Done.

L23-29: This sentence is too long. It is better to separate into two or more sentences.

Done.

L24: For “NR4+”, subscript and superscript are used.

Done.

L34: It is better to change “In the present work” into “In this review”.

Done.

L80: For the abbreviation of “logP”, more explanation is required.

Done.

L113: “cells, therefore” is changed to “cells. Therefore”.

Done.

L133: A compound “1c” is not shown in any scheme.

It has been corrected.

L154: What does mean the numeral “1”?

The numbers have been removed.

L173: It is better to change “form occurs” into “form is formed”.

Done.

L181: “AMP” is changed to “AMPs”.

Done.

L282: The phase of “18b able to inhibit” is modified.

Done.

L314: “in Table 1. [41].” Is changed to “in Table 1 [41].”

Done.

L331: “et al” is changed to “et al.”.

Done.

L423: “, however, it did not” is changed to “. However, it did not”.

Done.

L439: In this manuscript, the terms of review, work, and study are mixed. It is better to unify the use of term, for example, “review”.

Done.

L492: “et. al” is changed to “et al.”.

Done.

L498-500: This sentence should be modified for readers to enhance the understanding.

The sentence has been modified.

L539: “42a, however, due to” is changed to “42a. However, due to”.

Done.

L577: “GS were” is changed to “GSs were” or “GS was”.

Done.

L605: Is “GSs” correct?

Yes, it was corrected.

L665: “Table 2 and 4” is changed to “Tables 2 and 4”.

Done.

L704: The size of Scheme 35 is a little expanded.

The size has been adjusted.

L745: “This research team” Does it refer to which team?

It has been specified with “The research team led by Brycki”

L761: “Table 1 and 3” is changed to “Tables 1 and 3”.

Done.

L767: “These environmentally friendly molecules” is better.

Done.

L793: “Table 1 and 3” is changed to “Tables 1 and 3”.

Done.

L805: “[94, 95]” is changed to “[94,95]. Remove a space between 94, and 95.

Done.

L883: “hydroalocoholic solution” is changed to “hydroalocoholic solutions”.

Done.

L893: What was irradiated?

“hydro-alcoholic solution” was added.

L940: The size of Scheme 45 is a little expanded.

Scheme 45 has been resized.

L1004: There are two spaces between fragments and proved. One space is removed.

Done.

L1020: “In the reaction” is changed to “In this reaction”.

Done.

L1073: “Table 2, 4” is changed “Tables 2 and 4”.

Done.

L1101: The term “Table 4” is crosslined.

Done.

L1121: There are two spaces between The and congener. One space is removed.

Done.

L1126: The abbreviation of SEM is already used.

Done.

L1168: It is better to change “were reacted with” to “reacted with”.

Done.

L1201: “MMP” is changed to “MMPs”.

Done.

L1205: The abbreviation “IC50” is already used.

Done.

L1221: “HAP” is changed to “HAPs”.

Done.

L1404: It is better to change “were reacted with” to “reacted with”.

Done.

L1457: “but also demonstrated” is changed to “but also as demonstrated”.

Done.

L1555: “This molecule” is changed to “This compound”.

Done.

L1621: Here is “min”. The term “minute(s) is also used. In the term “hour(s) is also used. The unit of time should be unified.

Done.

L2041: Only this journal name is abbreviated.

All common abbreviations have been introduced.

Reviewer 2 Report

Comments and Suggestions for Authors

In this present review, Fedorowicz et al. aim to provide a succinct, comprehensive review of QAC entitled "Advances in the Synthesis of Biologically Active Quaternary Ammonium Compounds." Overall, the review looks good. Although the overall quality of the review is commendable, it should address a number of suggestions, and I propose substantial improvements to the article prior to proceeding with further evaluation.

·       The abstract is poorly written, and the language is difficult to understand. The authors should have explained why this review is significant in the field of QAC and how it differs from previous literature reviews. E.g., “The discourse explores the versatile applications of quaternary ammonium muscle relaxants and cholinesterase inhibitors.” I would recommend using “quaternary ammonium containing cholinesterase inhibitors and muscle relaxants.”

·       Need reference for the following line: “They are utilized in a variety of industrial and consumer products in chemistry (phase-transfer catalysts, ionic liquids, and low-vapor pressure solvents), agriculture (herbicides and pesticides), as well as in households (personal care products, sanitizers and cleansers, fabric softeners, and antistatic shampoos).”

·       NR4+ should be changed to NR4+

·       In the present line (34-36), the authors mentioned, “In the present work, the latest reports on the development of small-molecule QAC synthetic procedures as well as their bioactivity are presented.” It's worth mentioning in the review what time span the authors covered.

·       It is important to write a paragraph on “The structure of quaternary ammonium compounds.”

·       Line 44–47 reference is missing; provide a reference.

·       In lines 48–66, the authors should mention the references to the corresponding lines mentioned in the review. Merely writing a full paragraph and adding a bunch of references would not help the readers.

·       Line 87: selected newly synthesized QACs; how newly synthesized are the compounds? It is ideal to write “minimum inhibitory concentration (MIC) of the synthesized QACs found in the literature.”

·       Include a few lines and properly explain the effectiveness of synthesized quaternary ammonium salts differing in chain lengths.

·       This review article is on the synthesis of various important biologically active QACs, but it is always recommended to include “toxicity of QACs” in a separate paragraph to give the readers a clear idea of how to use the QACs to minimize their toxicity.

·       Mention “Present and Prospective Trends in the Application of QACs.”

·       Instead of using "antibacterial activity I would recommend using “antimicrobial activity” and add antibacterial, Antivirals, Antiprotozoals in subheading.

·       The authors mentioned in the title “…synthesis of biologically active quaternary ammonium compounds” From a chemistry perspective, it is really important to state the yield of the compounds presented here.

·       Some of the reaction conditions (e.g., temperature, time) are missing. Mention it in each scheme.

·       I would recommend including a section on “Abbreviation.”

·     In Scheme 21, the quinuclidine color is blue. I am curious to know what the blue color signifies. The same goes with scheme 22: DABCO. 

Comments on the Quality of English Language

The English in the abstract section should be rewritten. Overall, the writing is fine but needs minor English refinement.

Author Response

We thank the reviewer for the detailed inspections, which have led to significant improvements of the manuscript.

Reviewer 2

In this present review, Fedorowicz et al. aim to provide a succinct, comprehensive review of QAC entitled "Advances in the Synthesis of Biologically Active Quaternary Ammonium Compounds." Overall, the review looks good. Although the overall quality of the review is commendable, it should address a number of suggestions, and I propose substantial improvements to the article prior to proceeding with further evaluation.

1.·       The abstract is poorly written, and the language is difficult to understand. The authors should have explained why this review is significant in the field of QAC and how it differs from previous literature reviews. E.g., “The discourse explores the versatile applications of quaternary ammonium muscle relaxants and cholinesterase inhibitors.” I would recommend using “quaternary ammonium containing cholinesterase inhibitors and muscle relaxants.”

The abstract has rewritten.

2.·       Need reference for the following line: “They are utilized in a variety of industrial and consumer products in chemistry (phase-transfer catalysts, ionic liquids, and low-vapor pressure solvents), agriculture (herbicides and pesticides), as well as in households (personal care products, sanitizers and cleansers, fabric softeners, and antistatic shampoos).”

Done.

3,·       NR4+ should be changed to NR4+

Done.

4.·       In the present line (34-36), the authors mentioned, “In the present work, the latest reports on the development of small-molecule QAC synthetic procedures as well as their bioactivity are presented.” It's worth mentioning in the review what time span the authors covered.

Done.

5.·       It is important to write a paragraph on “The structure of quaternary ammonium compounds.”

Done.

6.·       Line 44–47 reference is missing; provide a reference.

Done.

7.·       In lines 48–66, the authors should mention the references to the corresponding lines mentioned in the review. Merely writing a full paragraph and adding a bunch of references would not help the readers.

Done.

8.·       Line 87: selected newly synthesized QACs; how newly synthesized are the compounds? It is ideal to write “minimum inhibitory concentration (MIC) of the synthesized QACs found in the literature.”

Done.

9.·       Include a few lines and properly explain the effectiveness of synthesized quaternary ammonium salts differing in chain lengths.

Done.

10.·       This review article is on the synthesis of various important biologically active QACs, but it is always recommended to include “toxicity of QACs” in a separate paragraph to give the readers a clear idea of how to use the QACs to minimize their toxicity.

Done.

11.·       Mention “Present and Prospective Trends in the Application of QACs.”

Done.

12.·       The authors mentioned in the title “…synthesis of biologically active quaternary ammonium compounds” From a chemistry perspective, it is really important to state the yield of the compounds presented here.

Done, however in certain papers yields have not been reported.

13.·       Some of the reaction conditions (e.g., temperature, time) are missing. Mention it in each scheme.

Done.

14.·       I would recommend including a section on “Abbreviation.”

Done.

15.·     In Scheme 21, the quinuclidine color is blue. I am curious to know what the blue color signifies. The same goes with scheme 22: DABCO.

The blue color along with descriptions ‘quinuclidine’ and ‘DABCO’ have been removed.

Round 2

Reviewer 2 Report

Comments and Suggestions for Authors

The manuscript “Advances in the synthesis of biologically active quaternary ammonium compounds”, authored by Fedorowicz et al., describes the recent advancements in the design and synthesis of biologically active quaternary ammonium compounds (QACs). Overall, this is a nice piece of work. The manuscript is well written, and the authors modified the manuscript according to the suggestions. From my perspective, the manuscript is suitable for acceptance in its present state.